# A multifaceted analysis of OTUD5 integrated MAVS in innate immunity of Primary Biliary Cholangitis

Ran Chen[1], Yan Sun [2]*, Wenlin Tai[1]*

1 Clinical laboratory, The Second Affiliated Hospital of Kunming Medical University, Kunming Yunnan, P.R. China, 2 Pharmaceutical College & Key Laboratory of Pharmacology for Natural Products of Yunnan Province, Kunming Medical University, Kunming Yunnan, P.R. China

* taiwenlin@kmmu.edu.cn (WT), 553046530@qq.com (YS)

## Abstract

### Background

Primary biliary cholangitis (PBC) is a chronic autoimmune liver disease characterized by cholestasis caused by intrahepatic small bile duct injury. Promoting the molecular mechanism of OTU deubiquitinase 5 (OTUD5) in the treatment of PBC requires further exploration. This study unraveled the molecular underpinnings of PBC through bio-informatics analysis and experimental verification for the development of targeted therapeutic strategies.

### Methods

This study screened immune-related genes and validated their expression patterns in whole blood of patients with PBC using microarray based on GEO datasets. The expression level of OTU deubiquitinase 5 (OTUD5) was validated in peripheral blood samples using RT-qPCR and immunofluorescence. Subsequently, proteomic bioinformatics analyses were conducted utilizing STRING and InBio Discover databases to predict interactions with the mitochondrial antiviral signalling protein (MAVS). Furthermore, immunochemical and immunofluorescence analyses of MAVS expression in liver tissues were conducted with a thorough analysis of immune cell infiltration specific to the disease by utilizing single-cell RNA sequencing (scRNA-seq) technology in peripheral blood mononuclear cells (PBMCs) derived from both patients with PBC and healthy controls.

### Results

Compared with those of healthy controls, the liver tissues of patients with PBC presented increased NK cell activation, monocyte/mast cell numbers, and eosinophil numbers. Compared with those in 10 healthy controls, the expression of OTUD5 and

---

**Data availability statement:** The datasets of sequencing are submitted in the BioProject database repository, no. PRJNA1289408 and the other datasets in the Mendeley Data, V1, doi: 10.17632/ps7nn5sdy9.1.

**Funding:** This study was supported by the National Natural Science Foundation of China 82560413 to WLT, the Joint Foundation of Department of Science and Technology of Yunnan Province 202501AY070001-248 to RC/202401AY070001-091 to YS, the Yunnan Provincial Department of Education Science Research Fund Project 2023J0256 to RC, the Investigator Initiated Trail Projects of the Second Affiliated Hospital of Kunming Medical University Project ynIIT2023018 to RC, and the Applied Basic Research Foundation of Yunnan Province 202501AT070594 to YS. The above institutions did not participate in the design of the study, collection, analysis, interpretation of data, or in writing the manuscript.

**Competing interests:** The authors have declared that no competing interests exist.

**Abbreviations:** CTLA-4, Cytotoxic T lymphocyte antigen 4; DEGs, Different expressed genes; GWAS, Genome wide association study; GSVA, Gene set Variation analysis; HLA, Human leukocyte antigen; Ichip, its derivative Immunochip; MAVS, Mitochondrial antiviral signaling protein; OTUD5, OTU deubiquitinase 5; PBC, Primary biliary cholangitis; PBMCs, Peripheral blood mononuclear cells; scRNA-seq, single-cell RNA sequencing; TNF, Tumor necrosis factor; UDCA, Ursodeoxycholic acid; XCI, X chromosome inactivation.

MAVS was increased in 16 tissues of patients with PBC. High expression of OTUD5-MAVS in subpopulation 11 mononuclear macrophages was screened by PBMC scRNA-seq, and mononuclear cells with the subgroup 11 phenotype presented highly differentiated characteristics. The expression of OTUD5 and MAVS was inhibited in RAW264.7 cells when OTUD5 was knocked down (P < 0.05).

## Conclusion

This study focused on the overexpression of OTUD5 and its interaction with MAVS within macrophage subset 11 in patients with primary biliary cholangitis (PBC). The expression of OTUD5 and MAVS is increased in patients with PBC and is a potential target for the diagnosis and treatment of PBC.

## 1. Introduction

Primary biliary cholangitis (PBC) is a chronic autoimmune liver disease characterized by cholestasis caused by intrahepatic small bile duct injury. It usually occurs in middle-

aged women and is characterized by positive serum antimitochondrial antibodies and increased autoreactive T and B cells [1]. In recent years, the annual global incidence and prevalence rates were 17.6 per million and 146 per million, respectively [2]. The reported incidence and prevalence of PBC in Asia Pacific (8.4 per million and 98.2–118.8 per million, respectively) are lower than those reported in North America (27.5 per million and 218.1 per million, respectively) and Europe (18.6 per million and 145.9 per million, respectively) [3,4]. There are also regional differences in prevalence in Asia Pacific, with higher prevalence rates in Japan and China (191.18 per million) [4]. Ursodeoxycholic acid (UDCA) and obeticholic acid are widely used as first-line drugs. Treatment methods such as those utilizing OCA, fibrin, peroxisome proliferator-activated receptor agonists, reduced nicotinamide adenine dinucleotide phosphate oxidase inhibitors, immunomodulators and mesenchymal stem cell transplantation can delay disease progression [5,6]. However, patients may experience asymptomatic, slowly progressing, symptomatic, or rapidly progressing stages of the disease.

Early genetic studies of PBC focused on human leucocyte antigen (HLA) and non-HLA candidate genes associated with necrosis factor. These include tumour necrosis factor (TNF), cytotoxic T lymphocyte antigen 4 (CTLA-4), toll-like receptors and vitamin D receptors [7–9]. Presently, genome-wide association studies (GWAS) and their derivative immunochip (iCHIP) association have gradually replaced these earlier studies. Large sample banks have been established in North America, Europe, Japan and China to facilitate high-throughput genetic studies of PBC [10]. Unlike GWAS, they are mainly involved in the secretion of IF-12 (IRF5, SOCS1, TNFAIP3, NF-B, and IL-12A), activation of T cells to produce IFN-γ (TNFSF15, IL12RTYK2, STAT4, SOCS1, TNFSF15, NF-κB, and TNFAIP3), and activation of B cells to produce immunoglobulins (POU2AF1, SPIB, PRKCB, IKZF3 and ARID3A)

[11–13]. However, significant differences in sex have been observed in cohort studies of the epidemiology, natural history, and clinical characteristics of patients with PBC, with the female/male incidence ratio recently reaching 9:1 [14]. The mechanism of epigenetic X chromosome inactivation (XCI) was avoided [15]. In addition, the proportion of the X haplotype in peripheral blood mononuclear cells of female patients with PBC increases with age [16], revealing the underlying immune regulatory mechanism of genes on chrX, which is of great significance for susceptible female groups. Notably, a chromosome X-wide common variant association study by Rosanna Asselta's group at the Center for Autoimmune Liver Disease of Medicine and Surgery of the University of Milan-Bicocca revealed that a unique linked imbalance block gene region on the X chromosome includes seven genes, namely, mitochondrial endometrial 17B (TIMM17B), polyglutamine-binding protein 1 (PQBP1), Pim-2 proto-oncogene serine/threonine kinase (PIM2), solute vector family 35 member A2 (SLC35A2), OTU deubiquitinase 5 (OTUD5), potassium voltage-gated channel subfamily D member 1 (KCND1) and Grip1-associated protein 1 (GRIPAP1), as well as a super enhancer GH0XJ048933 in PBC [17]. This finding appears to transcend the initial female-specific association reported in the original XWAS study by Asselta et al., which focused on a genetic locus and validated expression exclusively in female cohorts. However, the role of these genes in this region remain unknown, and the molecular mechanism still needs further validation.

This study measured the expression levels of OTUD5, PQBP1, and TIMM17B in patients with PBC and compared them with those observed in patients with PSC. Machine learning techniques were used to screen immune genes, and the target clusters were subsequently validated via a multisample whole-blood RNA microarray. We subsequently confirmed the overexpression of OTUD5 in patients with PBC by RT-qPCR. Potential interactions between OTUD5 and MAVS are suggested in proteomic predictions, offering intriguing insights into their potential interplay. Additionally, HSPA9 and TIMM17B are postulated to form signalling pathways involved in mitochondrial stress. To gain a deeper understanding of immune cell subsets in PBC, we conducted scRNA-seq on PBMC samples derived from patients with PBC and healthy controls. This comprehensive analysis revealed the presence of OTUD5-MAVS molecular markers in monocytes, along with significant variations in macrophage profiles in the peripheral blood of patients with PBC. While mitochondrial downstream signalling molecules associated with MAVS in PBC have yet to be reported, the activation of the MAVS signalling pathway during stages of innate immunity in response to pathogen infection triggers type I interferon expression. This response facilitates the clearance of invading pathogens through a nonspecific inflammatory response. In particular, an overactive type I interferon response can lead to immunopathological damage and potentially to autoimmune diseases. Investigating the regulatory mechanism between OTUD5 and MAVS in PBC offers a promising avenue for identifying novel intervention targets for PBC, providing new insights for clinical diagnosis and treatment strategies.

## 2. Materials and methods

### 2.1. Sample

All patients met the internationally accepted criteria for PBC. The RNA expression profiles of blood samples collected from 90 patients with PBC and 47 healthy controls and those of 45 PSC blood samples and 47 healthy controls from the GEO public database, time spans of data collection, location and setting, characteristics of the relevant population, and any inclusion or exclusion criteria used in the original study were determined. This study conformed to the ethical guidelines of the 1975 Declaration of Helsinki. The GEO data were approved by the local code in accordance with the protocol [18,19]. This study is approved by Human Experimental Ethical Committee of Kunming Medical University (No. PJ 2020−58).

### 2.2. Analysis method

Raw data of disease normalization and differential genes were screened. The SVM-RFE model was used to identify PBC characteristic gene sets, which were combined with LASSO regression model genes. Intersection verification and ROC analysis of clinical diagnostic efficacy were carried out. *CIBERSORT* was used to distinguish pathogenic genes

and PBC tissue-related immune cells and infiltration status. The GEO data were downloaded, annotated, merged, and screened for differentially expressed disease genes for use in a heatmap. Differential gene expression patterns between PBC samples were aligned. Differential genes were compared between PBC liver tissue (PBC-T) and whole blood (PBC-B) chips. A blood sample chip with a sufficient sample size was used, and PSC (PSC-B) whole blood chip data were added as the verification set for analysis of differential gene expression and immune cell infiltration status. CIBERSORT covers 22 common immune infiltration cell types. Differences in the cell distributions among the input samples were identified [20].

## 2.3. Differentially expressed genes (DEGs)

The target interactions of OTUD5 were predicted via inBio Discover and the STRING database. The DEGs in GSE79850 of PBC liver tissue were obtained. The GSE119600 chips from the GEO database were analysed to obtain DEGs between patients with PBC and healthy controls via the R software *limma*.

## 2.4. Immunohistological analysis

A total of 26 liver samples from patients with 16 PBC and 10 healthy controls were collected. The samples were fixed in 10% buffered formalin and sectioned at a thickness of 4 μm for immunohistochemical analysis. Deparaffinized and rehydrated sections were subjected to antigen retrieval in citrate buffer (pH 7.4) and blocked with 5% normal goat serum. The sections were incubated overnight at 4 °C with an anti-MAVS antibody (diluted 1:200; 14341−1-AP; Proteintech Co., CN) and then exposed to a secondary antibody (diluted 1:500; GB23303; Servicebio Co., CN). The expression of MAVS was analysed by HE staining, and the percentage of positive hepatocytes was graded by staining intensity (0–3 scale).

## 2.5. RT-qPCR

The primers used were as follows: OTUD5 F: 5'-GCCAGGTACAAGCAGTCAGT-3' and 5'-AGAGGGGGACATCTGCTGAA-3'.

## 2.6. Immunofluorescence analysis

Samples from five patients diagnosed with stage III PBC were collected, along with one liver tissue sample from a healthy volunteer serving as a normal control. Written informed consent was obtained from all patients or their legal guardians. The study protocol was approved by the hospital's Ethics Committee. OTUD5 and MAVS sections were incubated with antibodies against OTUD5 (diluted 1:200; PA5−20611, Thermo Fisher Ltd., USA) and MAVS (diluted 1:200; 14341−1-AP, Proteintech Co., CN). OTUD5 fluorescence signals were represented as green (FITC), MAVS signals appeared red (TRITC), and nuclei were stained blue (DAPI).

## 2.7. Single-cell sequencing

Single-cell sequencing was conducted using a Xanthobiologic's SeekOne® DD (Digital Droplet) Single-cell 3' Transcriptome Kit. This kit utilizes microfluidic technology to isolate and capture single cells in a water-in-oil environment, and then nucleic acid-modified bar-coded beads are used to molecularly tag RNA from different cell sources, ultimately creating high-throughput single-cell transcriptome libraries compatible with Illumina and MGI sequencers. Processing single-cell RNA sequencing data involves removing low-quality bases using a sliding window method, automatically detecting joint sequences, and discarding low-quality reads. The SeekSoul Tools pipeline was used to extract cell barcodes and UMI sequences, whereas STAR 2.5.1 was used for reference genome alignment. Feature counts were then used to count gene expression and select appropriate parameters based on sequencing chemistry and regions. The analysis steps

included standardization, feature selection, dimensionality reduction, clustering, and tSNE/UMAP visualization. Gene set variation analysis combined with CyroTRACE, Monocle3 Pseudotime, and other advanced cell differentiation analysis methods were subsequently performed.

### 2.8. Knockdown of OTUD5

To further investigate the function and underlying mechanism of OTUD5 in macrophages, vector recombinant shuttle plasmids and packaging plasmids were constructed to interfere with the expression of OTUD5. These included OTUD5-MUS-1170, OTUD5-MUS-944, OTUD5-MUS-1249, LV3-NC, and polybrene. The expression levels of OTUD5 and MAVS were quantitatively analysed using real-time fluorescent quantitative PCR and Western blot after virus transfection, and a stable transgenic mouse RAW264.7 macrophage line in which the OTUD5 gene was knocked down was established. Concurrently, immunoconfocal detection was performed on slides of stable cell lines expressing OTUD5 and MAVS to verify the expression and colocalization of both proteins following the construction and knockdown of the stable cell lines. Whole protein extracts (20 mg) were separated via 8% sodium dodecyl-sulfate polyacrylamide gel electrophoresis (SDS-PAGE) and electrophoretically transferred onto polyvinylamide fluoride membranes. After being blocked with 5% nonfat dry milk in Tris-buffered saline, the membranes were incubated overnight at 4 °C with rabbit polyclonal anti-OTUD5 (1:1,000, Cell Signaling Technology, Inc., Danvers, MA) and rabbit polyclonal anti-MAVS (1:1000, Proteintech, Wuhan, China), followed by incubation with a secondary horseradish peroxidase (HRP)-conjugated anti-rabbit IgG antibody (DakoCytomation, Glostrup, Denmark). Subsequently, specific bands were visualized using the ECL detection kit (GE Healthcare, Waukesha, WI, USA). Images were captured using a lumino-image analyser (LAS-4000; Fujifilm, Tokyo, Japan), and densitometry was performed using Multi Gauge software (Fujifilm, Japan).

### 2.9. Statistical analysis

Statistical significance was determined by one-way analysis of variance (*ANOVA*) with *Tukey's post hoc test* for multiple group comparisons (SPSS 27.0, USA), and Spearman's rank correlation for non-normally distributed continuous variables, assessed by Shapiro-Wilk test by GraphPad Prism 6, *P<0.05* indicated statistical significance.

## 3. Results

### 3.1. ML screened for differential gene hubs between PBC tissues and blood samples with DEGs of the GEO datasets

Including Nanostring transcript expression profile microarray data from 16 PBC and 8 normal control paraffin tissue samples in GSE79850. The risk of disease presentation was determined by the results of long-term follow-up, high risk (n = 9 requiring liver transplantation) and low risk (n = 7 fully effective for UDCA) and normal controls were identified, and their first liver biopsy was extracted. RNA was extracted and analyzed using nanostring transcriptomics successfully. Patients with progressive disease appear to have a distinct molecular feature. Materials with high risk (n = 9 requiring liver transplantation) and low risk (n = 7 fully responsive to UDCA) were treated together with a control liver with no lesions (n = 8). GSE119600 microarray used RNA expression profiles from 90 PBC and 47 healthy controls as the training set; PSC peripheral blood from RNA expression profiles from whole blood samples from 45 PSC and 47 healthy controls as a validation set (Fig 1). Comparative analysis of gene expression profiles of liver tissues between PBC patients and healthy controls in GSE79850, A total of 147 DEGs were obtained *(|log2FC|>1* and *FDR<0.05*), it included 80 up-regulated DEGs and 67 down-regulated DEGs (Supplementary 1A); expression analysis of whole blood between PBC patients and healthy controls, A total of 4446 DEGs were obtained, it contained 2369 up-regulated DEGs and 2077 down-regulated DEGs

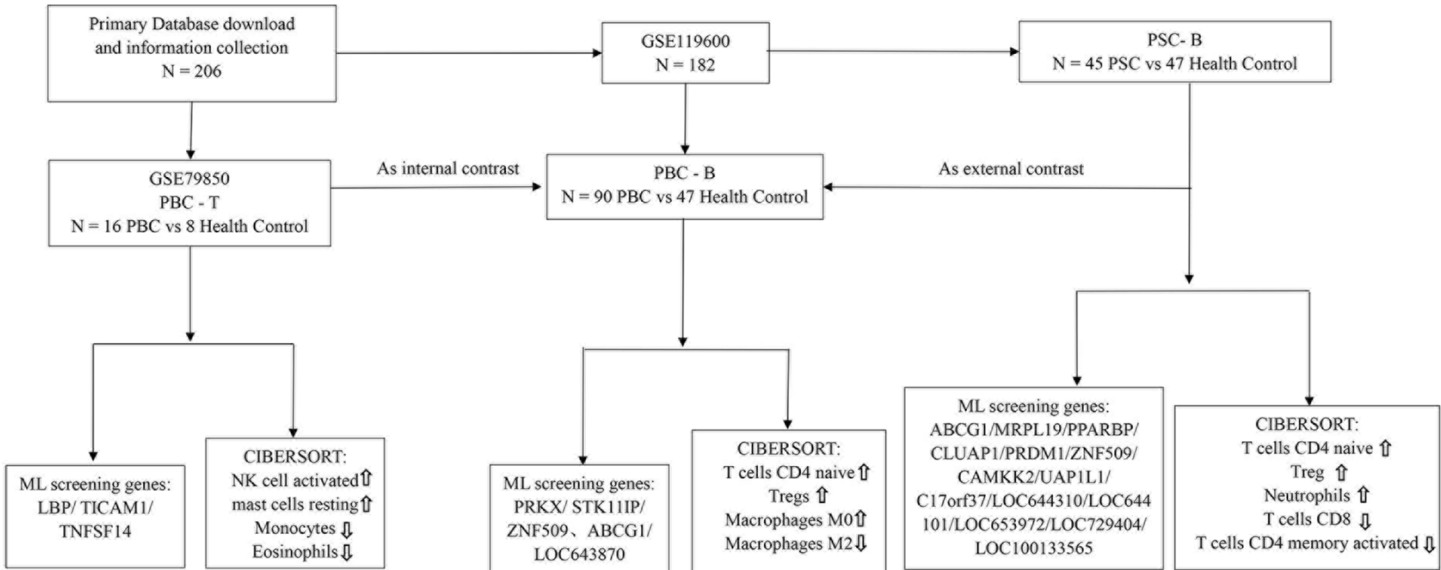

**Fig 1. Overall flowchart of this study.** The clinical tissues and blood samples in PBC and PSC were retained in the GEO database. After in-depth mining and analysis, different information of groups was found, then difference analysis, functional enrichment and machine learning screening were performed according to the above process, the correlation analysis of immune cells was performed finally. Differentially expressed genes (DEGs) hubs related to PBC in tissue samples. Heatmap showing significantly DEGs in PBC compared HC control. Using the RNA sequencing data of the GEO cohort to screen out (|log2FC|>1 and FDR<0.05). Log (λ) and the error model. The 2 dashed lines in Fig 2B indicate lambda.min and lambda.1se, respectively. Lambda.min denoted the value of λ when the model error was minimal. Lambda.1se denoted the model error within a standard error range of λ. SVM-REF approach to identify signature genes in PBC compared HC samples.Venn diagram based on the intersection of the two algorithms.

(Supplementary 1B); 555 DEGs differences in gene transcriptome expression profiles between PSC patients and healthy controls in GSE119600, it contains 170 up-regulated DEGs and 385 down-regulated DEGs Supplementary 1C).

### 3.2. Functional pathways and Distribution characteristics of peripheral blood mononuclear macrophage subpopulation in PBC

Top enrichments in biological process (BP) are ribonucleo protein complex biogenesis generation of precursor metabolites and energy, ncRNA processing, establishment of protein localization to organelle, mRNA processing, histone modification, positive regulation of cellular catabolic process, proteasomal protein catabolic process, positive regulation of cytokine production of GO enrichment analysis in PBC (Supplementary 2B). Cellular component (CC) are the mitochondrial inner membrane, mitochondrial matrix, mitochondrial protein-containing complex, cell-substrate junction, focal adhesion, nuclear envelope, transcription regulator complex, ribosome, actin cytoskeleton; Molecular function MF is the GTPase regulator activity, nucleoside-triphosphatase regulator activity, transcription coregulator activity, DNA-binding transcription factor binding, GTPase binding Pathway analysis: DEGs with differential expression differences were mainly enriched in positive cytokine regulation and cytokine receptor interaction pathway; GSEA analysis DEGs were mainly enriched in external encapsulating structure organization, peptidyltyrosine modification, regulation of immune effector process (Fig 2). Liver is one of the organs densely packed with many types of immune cells. In our study, it showed that the tissues of PBC showed NK cells activated, Monocytes, Mast cells resting, Eosinophilsand contrast to HC group obviously (Fig 3A,3B). Peripheral blood analysis of PBC exhibited T lymphocyte and macrophage compared to the control group.

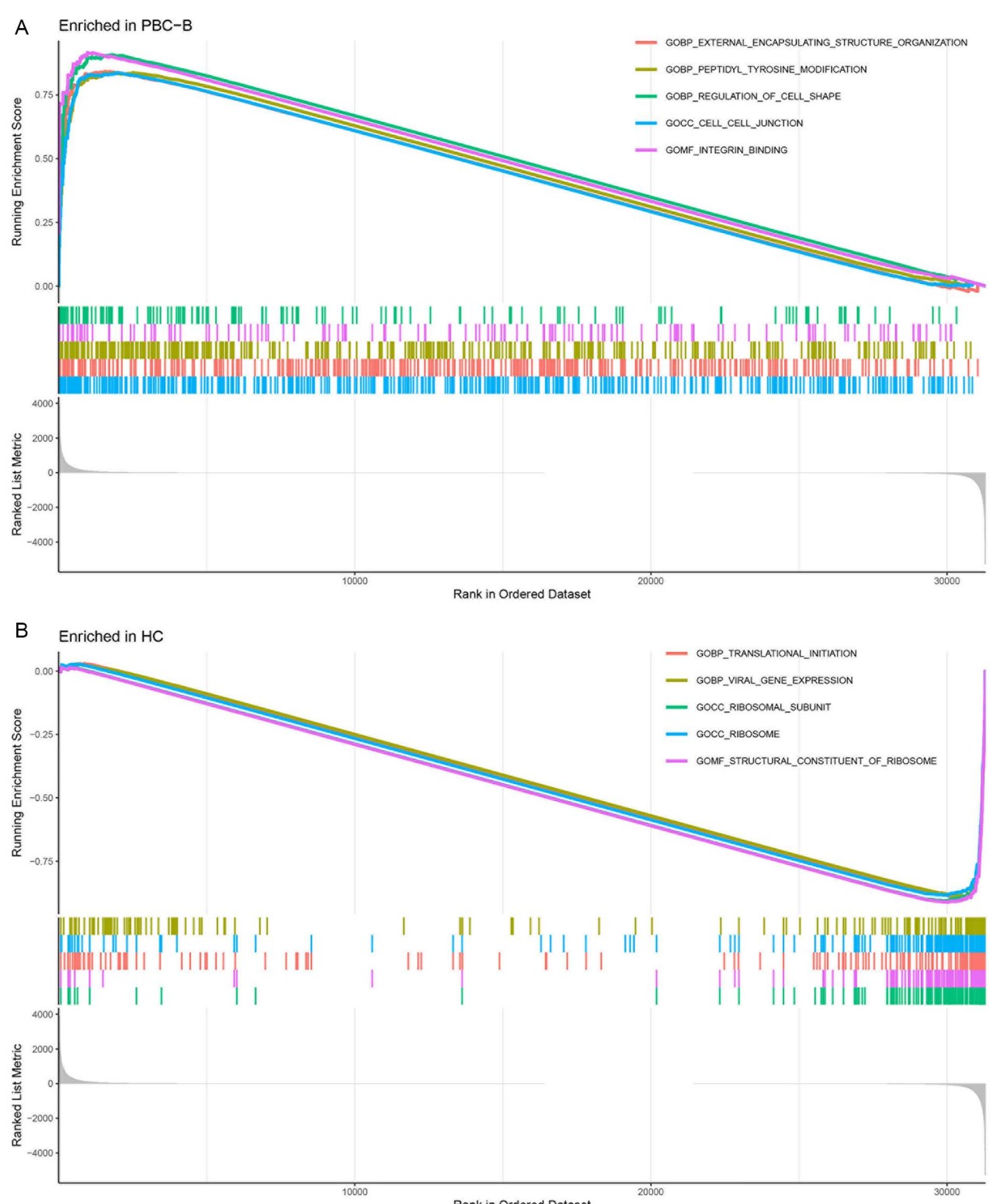

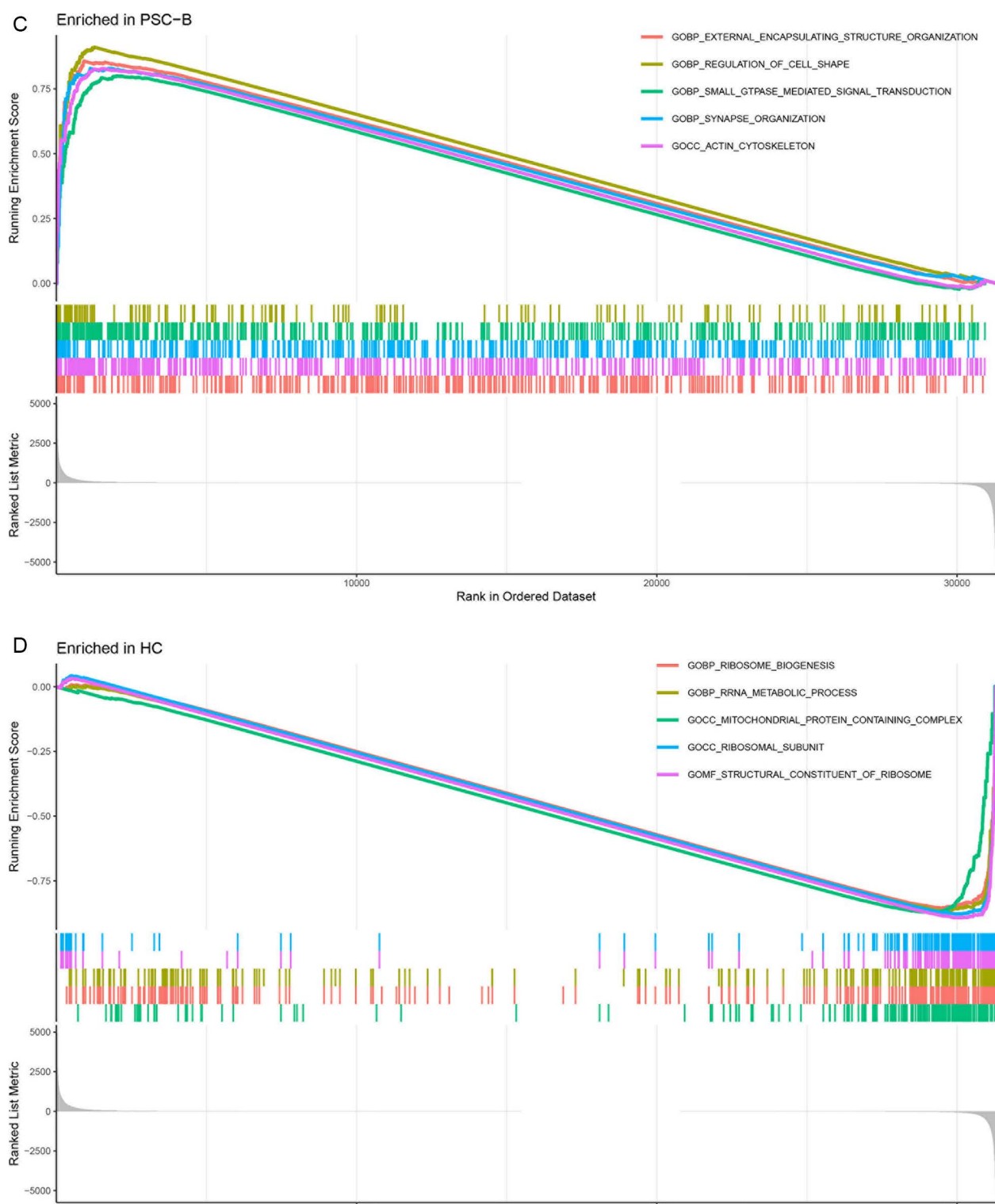

**Fig 2. GSEA results showing pathways enriched in the top or bottom of the ranked list of PBC and PSC in blood samples with the corresponding to enrichment in upregulated and downregulated genes (Note: The sample size of PBC tissue was limit so it couldn't be analyzed by GSEA). A.** Enrichment plot for the top 5 pathways in PBC-B. **B.** Top 5 pathways in HC control to PBC-B. **C.** Enrichment plot for the top 5 pathways in PSC-B. **D.** Top 5 pathways in HC control.

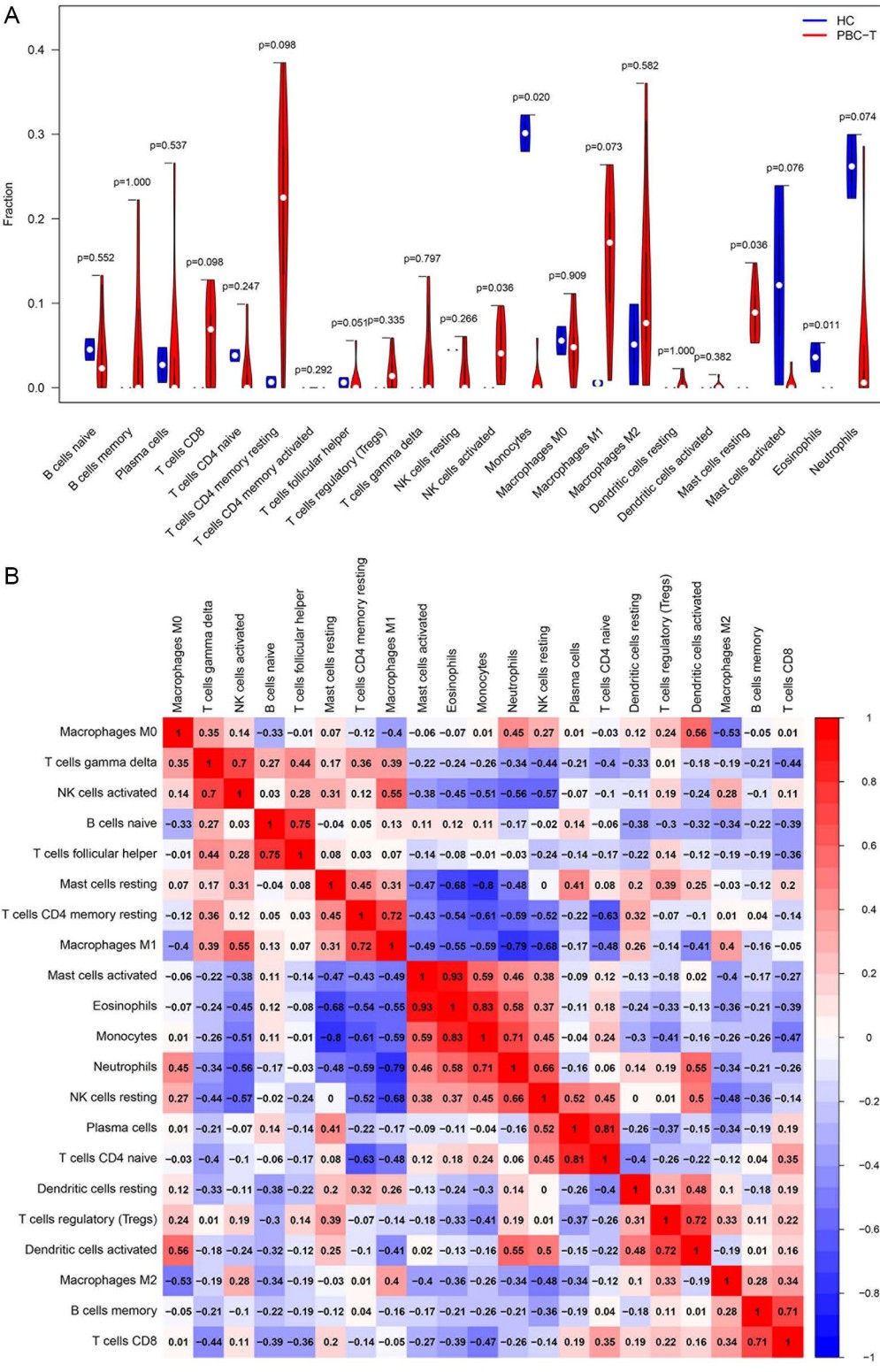

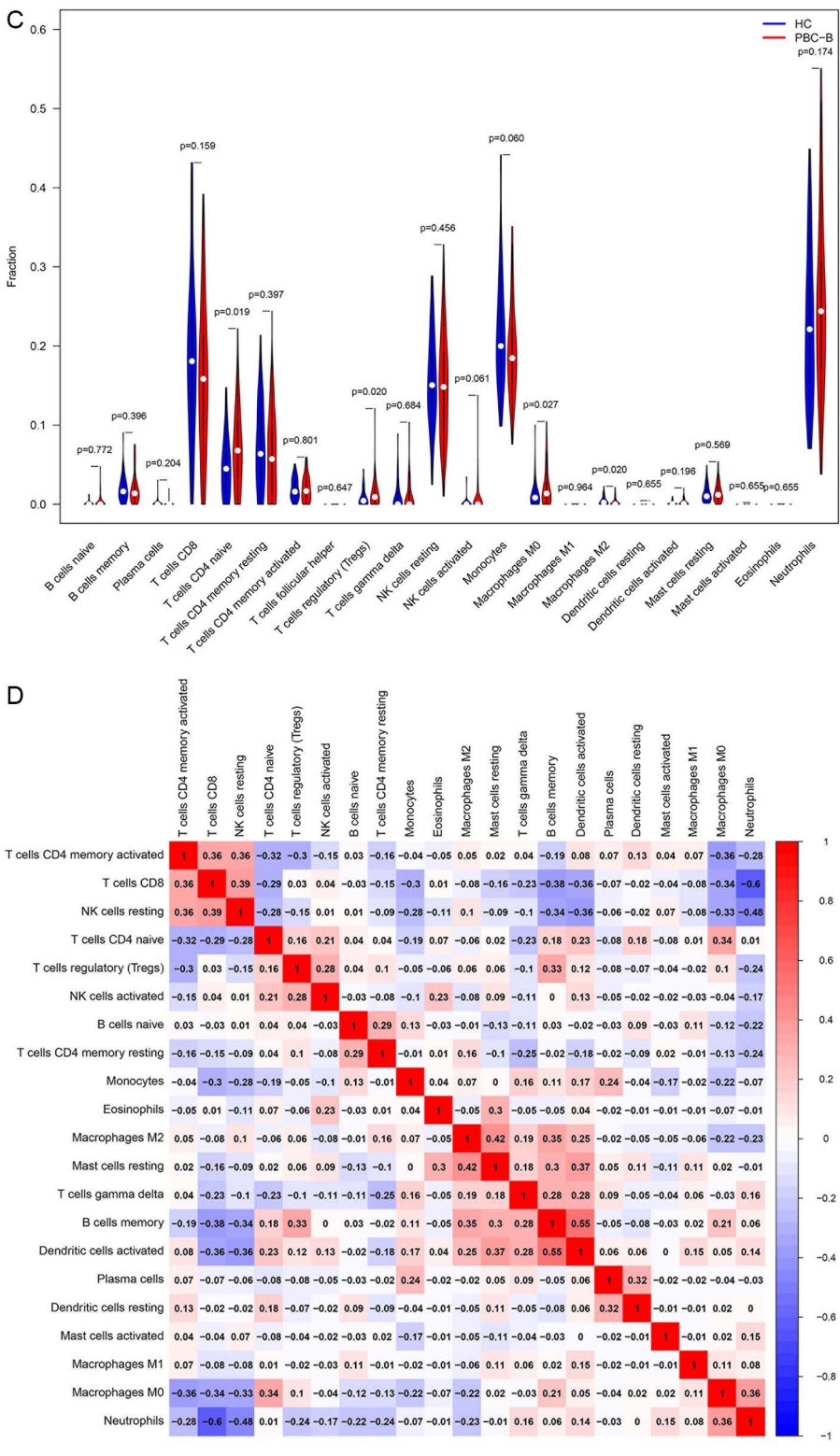

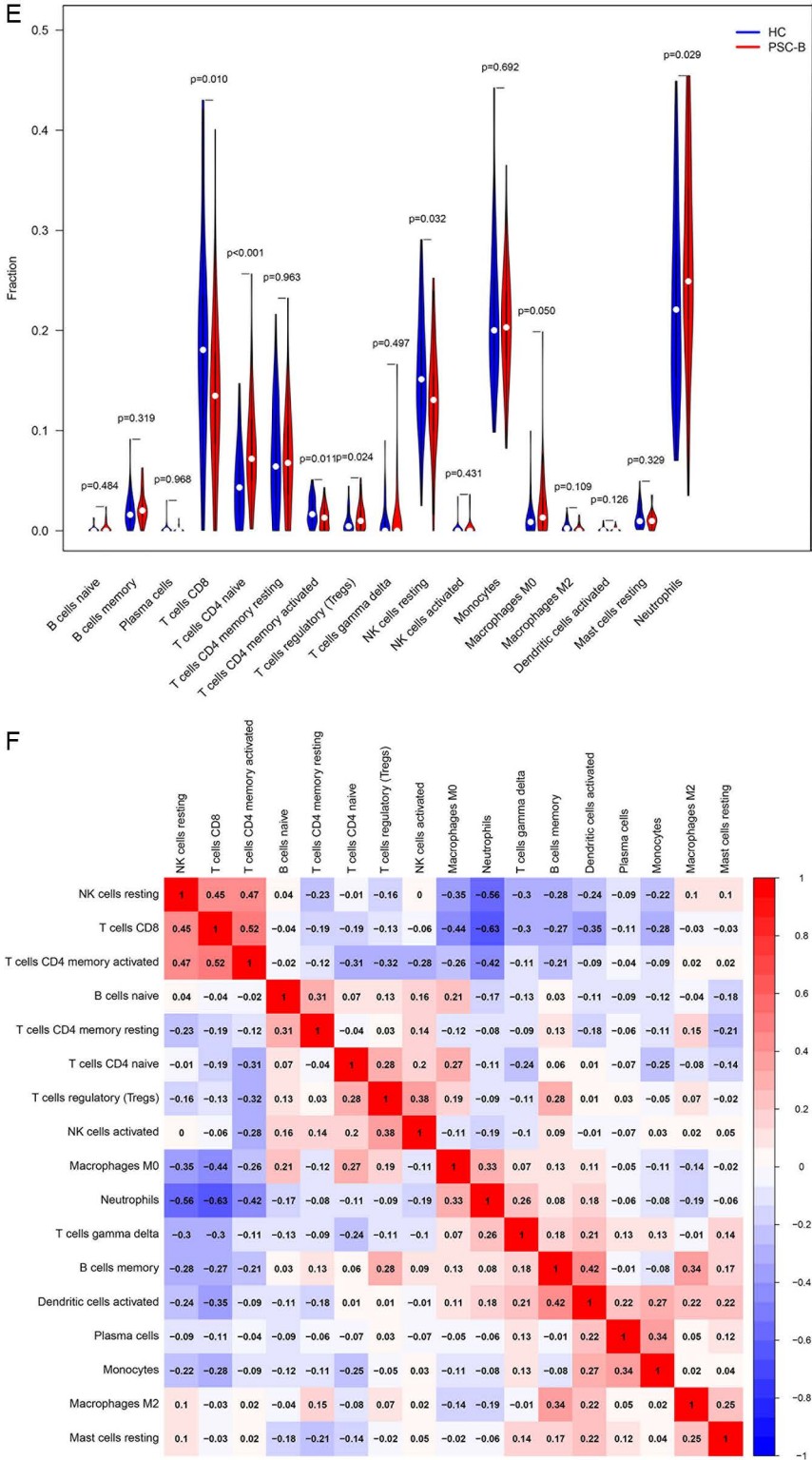

**Fig 3. The infiltration proportion of immune cell subsets in different tissues was significantly discrepancy between PBC and PSC. A.** Immune cell infiltration in tissue samples of PBC and HC control. The increase of NK cells activated, resting Mast cells, and decrease of Eosinophils and

Monocytes in PBC-T. **B.** Correlation analysis among various immune cells in PBC-T (The closer the absolute value of the correlation coefficient is to 1, the stronger the correlation between the variables). **C.** Immune cell infiltration in whole blood samples of PBC and HC control. The increase of CD4 naive T cells, Treg cells, and M0 macrophages, whereas M2 macrophage levels exhibited a significant decline in PBC-B. **D.** Correlation analysis among various immune cells in PBC-B. **E.** Immune cell infiltration in whole blood samples of PSC and HC control. The extreme increase of Neutrophiles, CD4 naive T cells, the decrease of CD8 T cells and resting NK cells in PSC-T. **F.** Correlation analysis among various immune cells in PSC-B.

Consequently, levels of CD4 naive T cells, Treg cells and macrophages M0 significantly increased, while macrophages M2 levels significantly decreased(Fig 3C,3D). Compared with PBC, the proportion of immune cell infiltration in whole blood of PSC patients was more different due to CD8 T cells, Neutrophils, NK cells resting, etc (Fig 3E,3F).

### 3.3. Expression level of OTUD5, PQBP1, TIMM17B, PIM2, SLC35A2, KCND1 and GRIPAP1 between PBC and PSC, MAVS expressed highly in tissue of microarray

After the intersection of SVM-REF and Lasso analysis, PBC-T identified LBP, TICAM1, and TNFSF14 as the selected gene sets, while PBC-B chose PRKX, STK11IP, ZNF509, ABCG1, and LOC643870. As an external reference, PSC-B selected a gene set comprising ABCG1, MRPL19, PPARBP, CLUAP1, PRDM1, ZNF509, CAMKK2, UAP1L1, C17orf37, and various LOC numbers including LOC644310, LOC644101, LOC653972, LOC729404, and LOC100133565.

Given that OTUD5, PQBP1, TIMM17B, PIM2, SLC35A2, KCND1 and GRIPAP1 are female sex-biased X chromosome genes, they may not be as highly expressed as other disease genes and were not prioritized by the aforementioned methods. However, their expression levels still exhibited statistically significant differences. Following the expression verification of PBC and PSC in blood samples from healthy individuals, it was observed that the expression of OTUD5 was significantly higher in PBC, whereas the expression of PQBP1 and TIMM17B was notably lower (Fig 4A–4C). In PSC, the expression of SLC35A2 and GRIPAP1 was significantly upregulated, while the expression of TIMM17B was downregulated (Fig 4D–4F).

It is worth noting the heightened expression of OTUD5 in PBC blood, to further delve into the molecular target of OTUD5, we meticulously selected MAVS as a potential target gene, drawing upon the bioinformatics of the proteins interaction network (Fig 5A–5C). The function anotation in inBio Discover database was confirmed in Table 1 . By integrating and thoroughly analyzing the raw chip data from 16 PBC cases and 8 healthy controls within the GEO database GSE79850 (Fig 6A), we successfully identified MAVS and observed its significantly elevated expression levels in 16 PBC tissues compared with 10 healthy controls (Fig 6B,6C).

### 3.4. The consistent over-expression of OTUD5 has been confirmed in numerous clinical peripheral blood samples, mirroring similar trends observed in pathological sections

By RT-qPCR, we have confirmed results from the peripheral blood of 26 hospitalized PBC patients and 22 healthy controls during the period from 2022 to 2023. A significant increase in the expression level of OTUD5 was observed.(Fig 7A). This finding is in precise agreement with the results observed from the over-expression of OTUD5 in PBC-B of our preceding research. Drawing from the network elucidated by bioinformatics analysis, which highlighted the targeting relationship between OTUD5 and MAVS, we subsequently gathered pathological paraffin sections of liver biopsy specimens from PBC stage I-IV patients at the same time.

Due to the correlation of OTUD5 with MAVS in the mononuclear macrophage system in immune cell of GEO analysis(Fig 7B,7C), It revealed a conspicuous upward trend in MAVS protein expression, aligning with the expression pattern observed in the GEO dataset. These specimens underwent immunohistochemical staining and confocal localization via

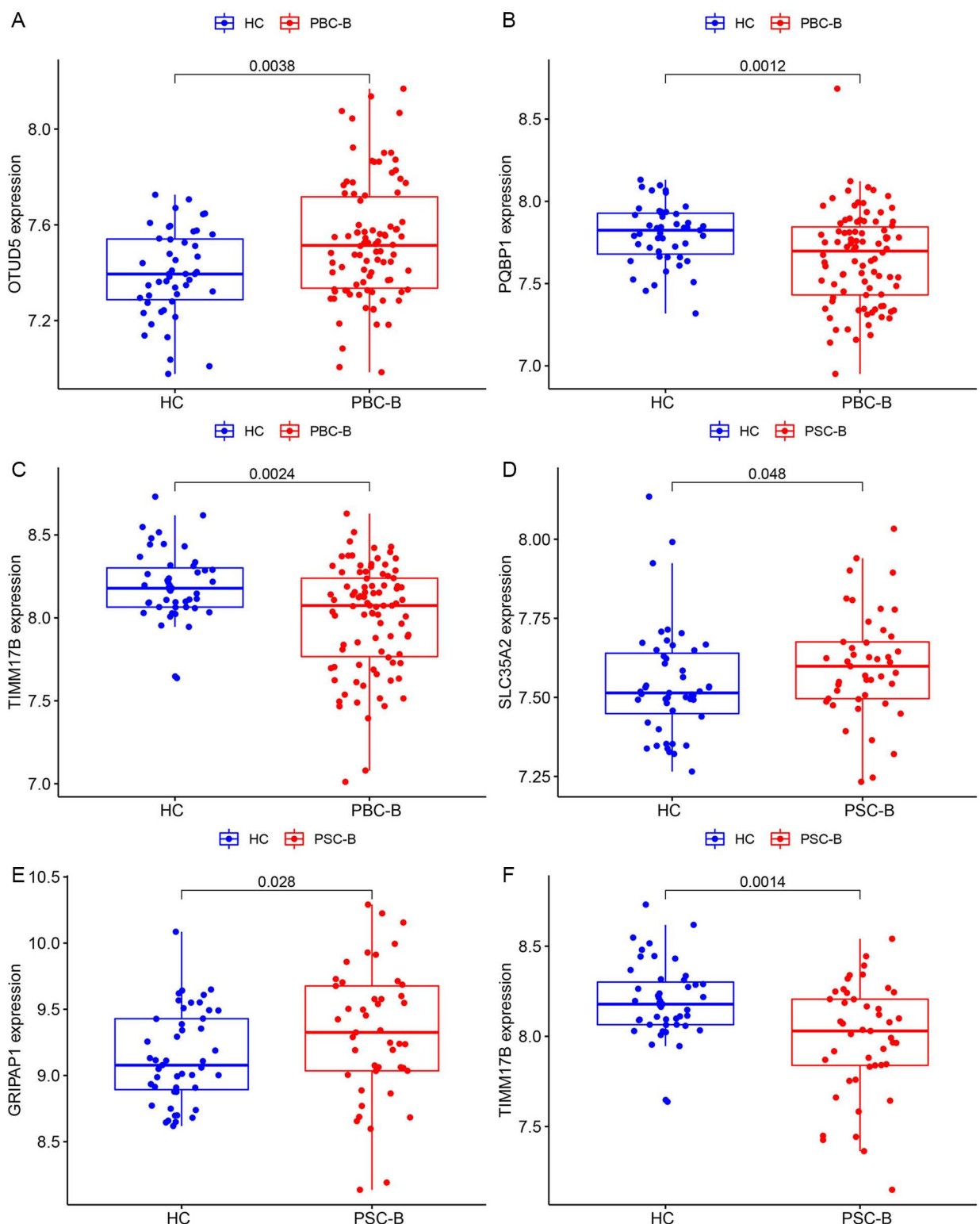

**Fig 4. The difference of X chromosome gene cluster expression in whole blood of PBC patients and PSC patients.** A&B&C. The expression of OTUD5, PQBP1, TIMM17B in the whole blood linked imbalance block gene region of PBC-B. D&E&F. Expression of SLC35A2, GRIPAP1, TIMM17B in PSC-B whole blood linkage imbalance block gene region. High-expression genes were marked with red markers, low-expression genes with green markers, and non-differential genes with black markers. P*<0.05.

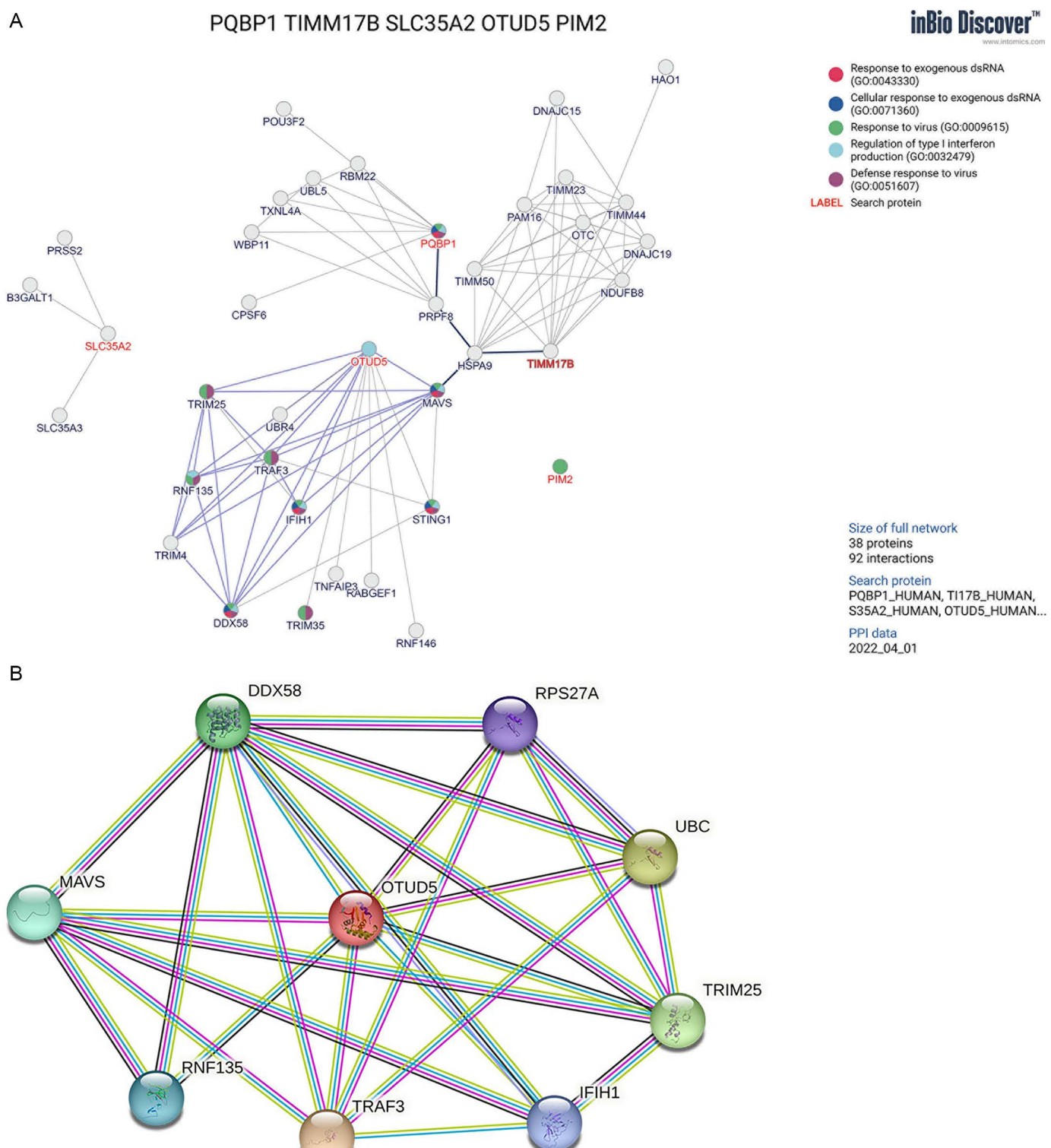

**Fig 5. Interaction between OTUD5 and MAVS in protein network prediction database. A.** The predicted results of protein interaction in inBio Discover database PQBP1/ OTUD5/ TIMM17B/ PIM2/ SLC35A2. **B.** The protein network of OTUD5 and MAVS in STRING database.

**Table 1. The predicted results of protein interaction in inBio Discover database PQBP1/ OTUD5/ TIMM17B/ PIM2/ SLC35A2.**

| Annotation | Size | Overlap | Ratio | P value |
|---|---|---|---|---|
| Response to exogenous dsRNA (GO:0043330) | 14 | 5/38 | >99.99 | 3.7e-11 |
| Cellular response to exogenous dsRNA (GO:0071360) | 14 | 5/38 | >99.99 | 3.7e-11 |
| Response to virus (GO:0009615) | 268 | 10/38 | 19.64 | 5.4e-11 |
| Regulation of type I interferon production (GO:0032479) | 73 | 7/38 | 50.47 | 7.4e-11 |
| Defense response to virus (GO:0051607) | 194 | 9/38 | 24.42 | 8.1e-11 |

three-label immunofluorescence(Fig 7D,7E). Leveraging immunofluorescence labeling technology on tissue slices from PBC patients, our research team definitively confirmed the substantial co-localization of OTUD5 and MAVS proteins. Significantly, this phenomenon was especially evident in the labeling sites of macrophages. Hence, we hypothesize that the OTUD5-MAVS complex may exert a pivotal regulatory influence in the progression of PBC disease, exhibiting tissue-specific significance.

### 3.5. High expression of OTUD5-MAVS in Subpopulation 11 of mononuclear macrophages was screened by PBMC ScRNA-seq

We collected peripheral blood mononuclear cells (PBMC) from 2 patients with primary biliary cirrhosis (PBC) and 2 healthy people, and performed single cell RNA sequencing (scRNA-seq) (Fig 8A,8B). According to the cell clustering rule of resolution 0.5_d30, we observed a significant increase in monocyte subpopulation 11 in PBC compared to healthy controls(Fig 8C). The expression of CD68 was the highest in monocyte subgroup 11, and the expression levels of OTUD5 and MAVS were significantly increased compared with other monocyte subgroups(Fig 8D). However, it is interesting to note that compared with MAVS, the number of cells differentiated from myeloid progenitor cells to monocytes in OTUD5 pseudotemporal distribution is highe (Fig 8E). Quasi-temporal analysis of the effect of cell differentiation time corresponding to OTUD5 showed that OTUD5 differentiated into monocytes in myeloid mesenchymal cells, and then gradually transformed into NK cells and B cells with the development of PBC (Fig 8F). Therefore, we advanced further analysis of subgroup 11 properties. This cell subpopulation has unique expression characteristics of IFI30, TCF7L2, AIF1, IFITM3, LST1 and other molecules(Fig 9A), and its functions are mainly focused on mitogen-activated protein kinase binding and GTPase activity(Fig 9B). Of particular note, M1 macrophage differentiation is mediated by Mitogen-Activated Protein Kinase binding and NF-κB signaling pathways. For human monocyte-derived macrophages, the enhanced migration capacity was influenced by GTPase activating proteins and regulatory protein kinases, which led to significant changes in the actin filament network.Given these characteristics, we can infer that this subpopulation is more similar to the M1 macrophage subpopulation. Subgroup 11 is terminal differentiated cells, which is a mature cell community that does not differentiate into B cells and NK cells alone. Through GSVA (gene set Variation analysis), we found that the expression of IFN-γ and IFN-α associated with inflammation was most significant in subgroups 11 further(Fig 9C). From the perspective of macrophage subpopulation differentiation, mononuclear cells with subgroup 11 phenotype have highly differentiated characteristics and are closely related to LST1 and TCF7L2 molecules(Fig 9D–9G).

In comparison with other mononuclear subgroups, such as subgroups 14, 15, and 21, the interactions between subgroups 11 and 14 were strongest(Fig 10A,10B). This is mainly reflected in the significant effects of CCR1 receptor and chemokines such as CCL3, CCL3L1, CCL5, as well as the strong effects of both on the regulatory signaling pathway of TGFβ1 and its receptor(Fig 10C,10D). After SCENIC predicted regulon of RSS analysis, we found that KLF4, KLF10, MAFB, POU2F2, RXRA transcription factors played specific regulatory roles in subgroup 11 of monocyte (Fig 10E–10I).In

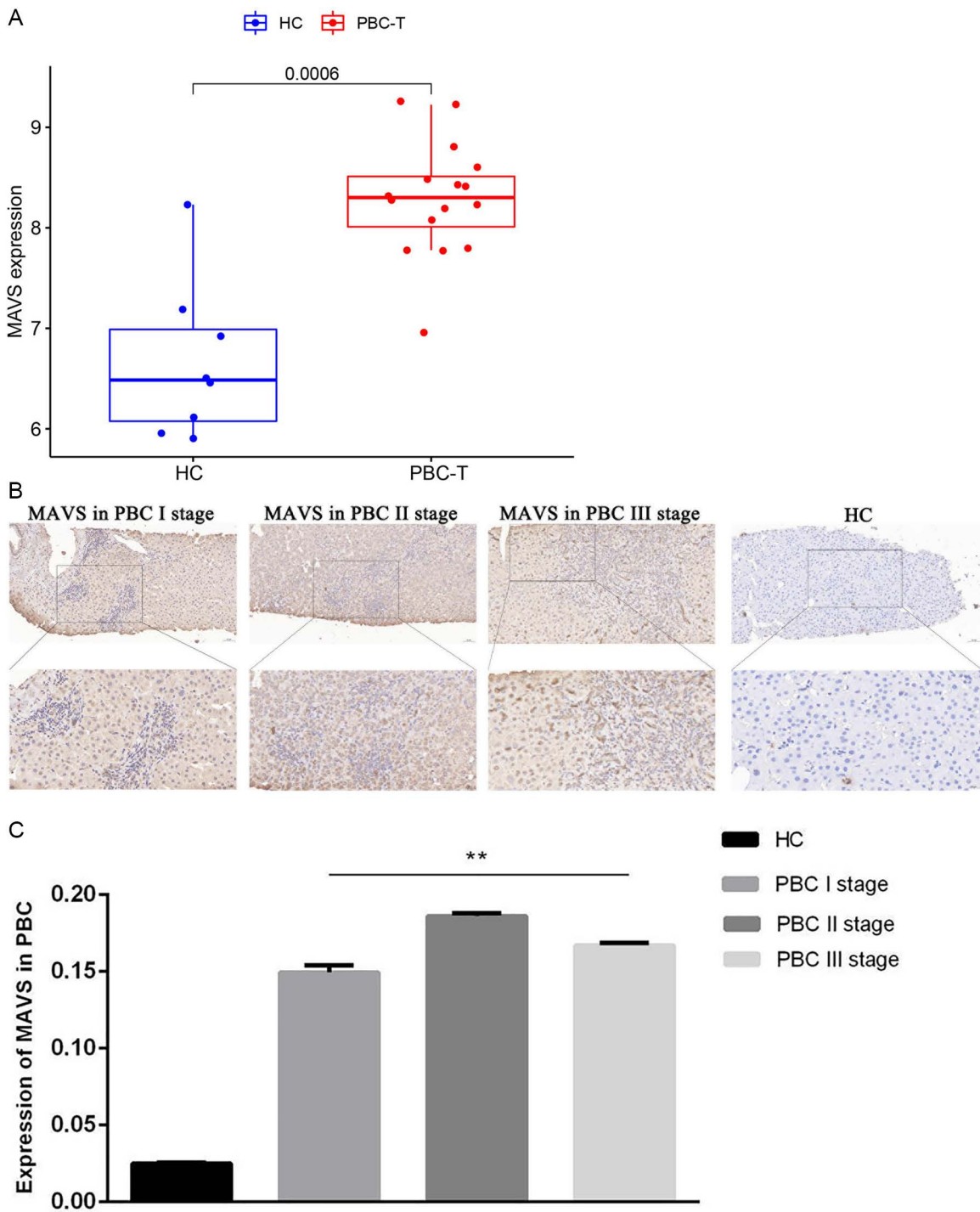

**Fig 6. Over-expression of MAVS in PBC-T and positive immunohistochemical staining within the liver tissue of clinical PBC patients. A.** MAVS demonstrated a marked increase in expression levels in the GEO analysis of PBC-T. **B.** The comparison of phase I-III immunohistochemistry between healthy controls and PBC patients. **C.** The statistical results of liver tissue immunohistochemistry. Scar bar = 50 µm in the upper Fig, and Scar bar = 20µm in local magnification in the lower Fig). P* < 0.05.

summary, this distinct subgroup 11 cell displayed analogous properties to M1 macrophages, a similarity that was validated not only in the NF-κB and TGF-β signaling pathways downstream of the MAVS molecule but also through the identification of characteristic catalytic and transcription factor action sites. These revelations offer crucial insights into the biological attributes of this cell subgroup.

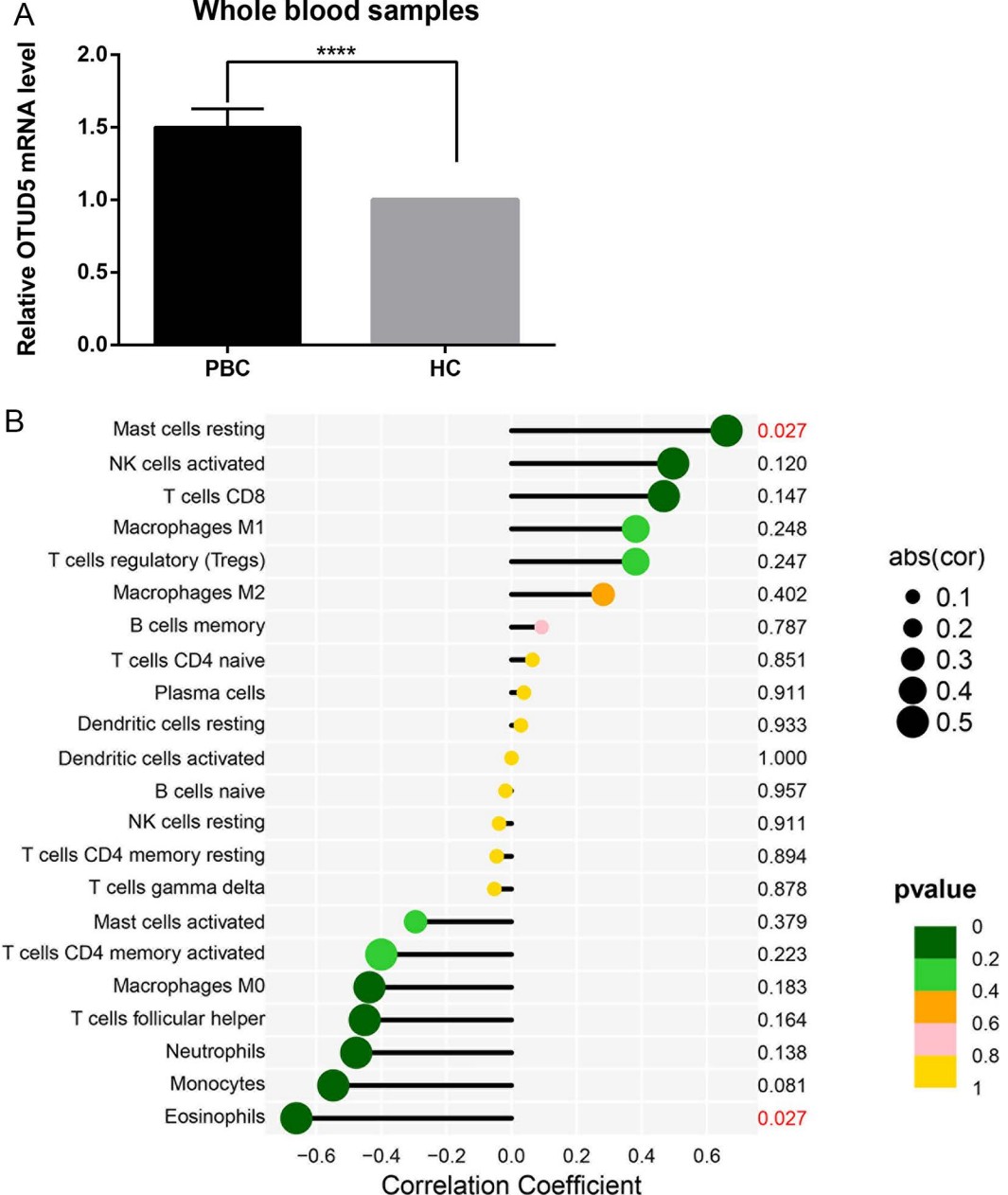

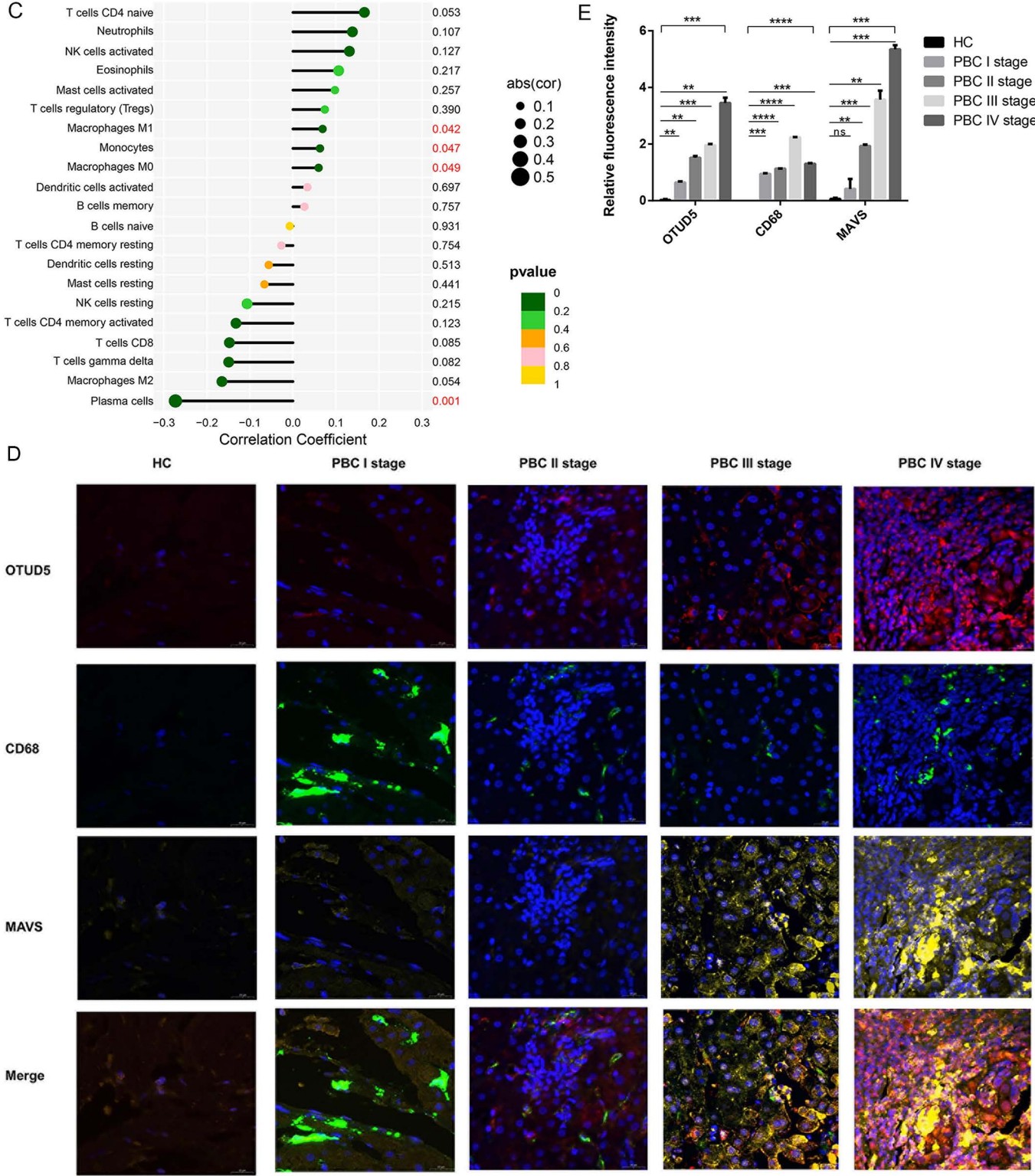

**Fig 7. Over-expression of OTUD5 in whole blood samples of PBC patients and its immunofocusing with MAVS in paraffin sections of liver tissue. A.** Following the execution of RT-qPCR detection and subsequent statistical analysis, it has been determined that the over-expression of the OTUD5 gene is statistically significant in whole blood samples from PBC patients relative to healthy control. **B.** MAVS is positively correlated with resting

mast cells and negatively correlated with eosinophils in PBC-T of GEO analysis. **C.** OTUD5 is positively correlated with monocyte/macrophage subsets and negatively correlated with plasma cells in PBC-B of GEO analysis. **D.** Over-expression of OTUD5 and MAVS in the immunofocusing of liver tissue was analyzed across PBC stages I-IV, respectively. The healthy control group served as the reference. Scale bar: 20 μm. **E.** The immunofocusing statistics of MAVS in liver tissues of PBC patients. P*<0.05.

### 3.6. Expression and fluorescence colocalization of MAVS in RAW264.7 macrophages after OTUD5 knockdown

The Otud5-Mus-1170, Otud5-Mus-1249, Otud5-Mus-944, and control LV3-NC were coated with lentivirus and transfected into RAW264.7 cells, achieving a transfection efficiency exceeding 70%. Subsequently, the protein from the transfected cells was extracted to verify the knockdown efficiency at the protein level. Compared to LV3-NC, the expression of OTUD5 in RAW264.7 was significantly inhibited at the protein level (OTUD5-MUS-1249 *P=0.0282*, OTUD5-MUS-944 *P=0.0058)*, as depicted in Fig 11A,11B. Additionally, the expression of MAVS in RAW264.7 was also significantly suppressed at the protein level, as shown in Fig 11C,11D (Otud5-Mus-944 *P=0.0389*, Otud5-Mus-1249 P=0.0221). To further confirm the cellular localization of the target protein MAVS and the specific regulatory mechanism of macrophages following knockdown, we conducted confocal immunofluorescence on Otud5-Mus-944, which exhibited the highest knockdown efficiency, and the LV3-NC unloaded control. The results indicated that after OTUD5 deletion, the fluorescence intensity of MAVS in RAW264.7 cells was weaker than that in LV3-NC (*P=0.0010,* Fig 11E,11G). Moreover, the quantified fluorescence expression in the OTUD5-MUS-944 group was significantly lower than in the blank-control, and MAVS was obviously down-regulated (*P=0.0179.* Fig 11F,11G).

## 4. Discussion

PBC is a chronic liver disease characterized by inflammation of the intrahepatic bile ducts, cholestasis, and immune cell infiltration [21]. Patients with PBC have high levels of immune cell infiltration. These immune cells induce the hepatic inflammatory response, exacerbate bile duct injury and vacuolation and lead to severe complications such as cirrhosis and portal hypertension [22]. Moreover, abnormal activation of B cells and antibody production play crucial roles in the pathogenesis of PBC. CD4$^m$ T cells secrete various cytokines and chemokines and attract other immune cells into inflammatory sites, which participate in the regulation of immune responses [23].

This study analysed PBC gene profiles and utilized machine learning algorithms to analyse the GSE 79850 and GSE 119600 datasets. After cross-validation via SVM-REF and *Lasso* analysis, the expression of LBP, TICAM1 and TNFSF14 was closely related to the immune response in patients with PBC-T. The expression of PRKX, STK11IP, ZNF509, ABCG1, and LOC643870 played a pivotal role in regulating immune cell functions in patients with PBC-B. Similarly, the enrichment of genes such as ABCG1, MRPL19, and PPARBP underscored the intricate nature of the immune network in those with PSC-B. Compared with the findings of other studies, the distinctive roles of LBP, TNFSF14 and ABCG1 in immune cell infiltration are highlighted [24–26]. The functional enrichment analysis of these genes revealed a profound link between immune cells and the inflammatory response. The liver is an organ densely populated by immune cells. This study found that NK cells were activated in PBC, whereas monocytes and mast cells remained in the resting phase. Compared with those in healthy controls, the ratios of T lymphocytes and macrophages, including increases in CD4 primary T cells, Treg cells, and M0 macrophages and a decrease in M2 macrophages, were altered in the peripheral blood of patients with PBC. PSC is associated with a different proportion of immune cells due to the shift in CD8+T cells, neutrophils, and resting NK cells [27–30].

This study analysed functional enrichment and immune cell infiltration to explore changes in PBC-related subpopulations of macrophages. The expression of OTUD5 increased in patients with PBC, while the expression of PQBP1 and TIMM17B decreased; the expression of SLC35A2 and GRIPAP1 decreased in patients with PSC. The expression of

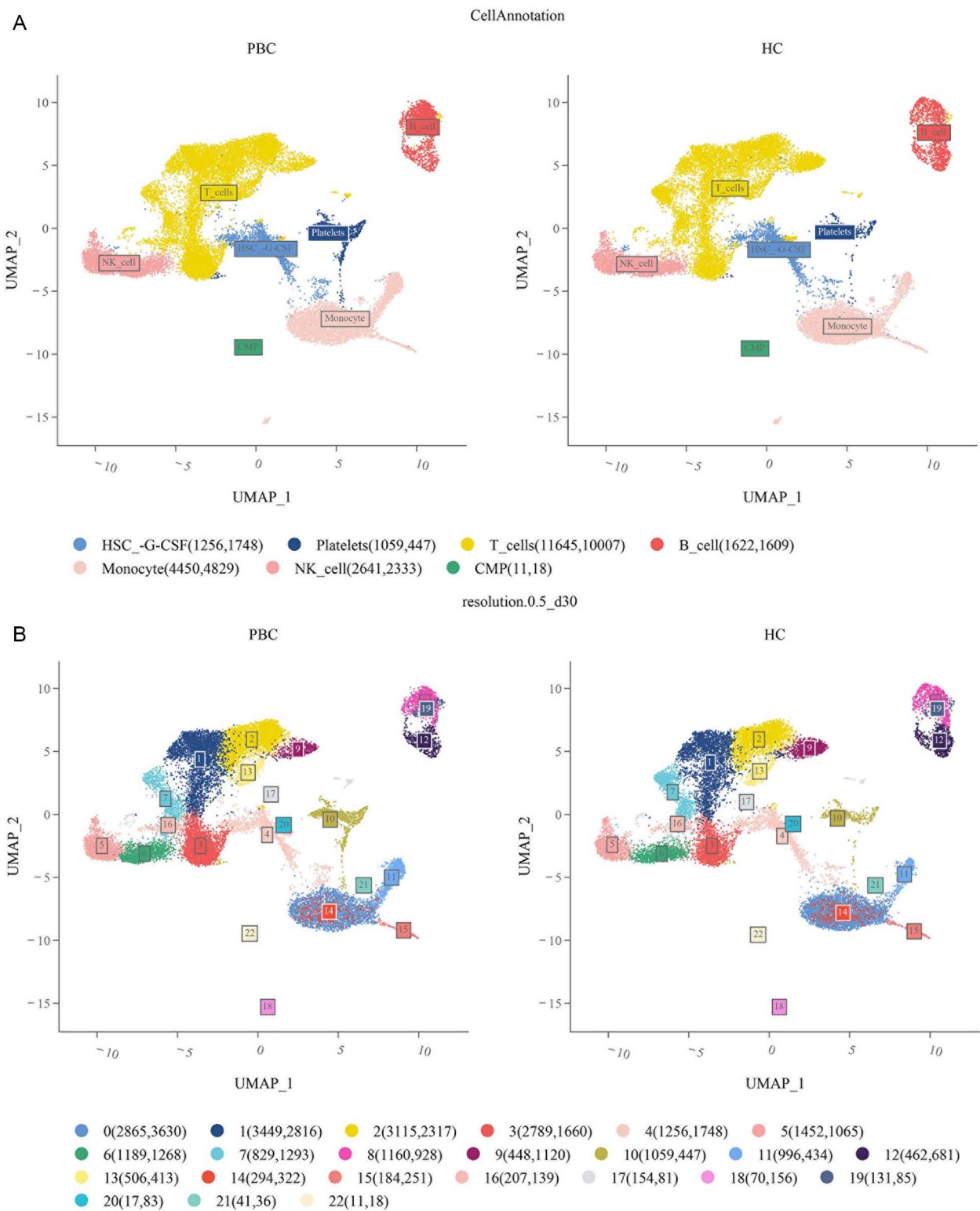

C

resolution.0.5_d30

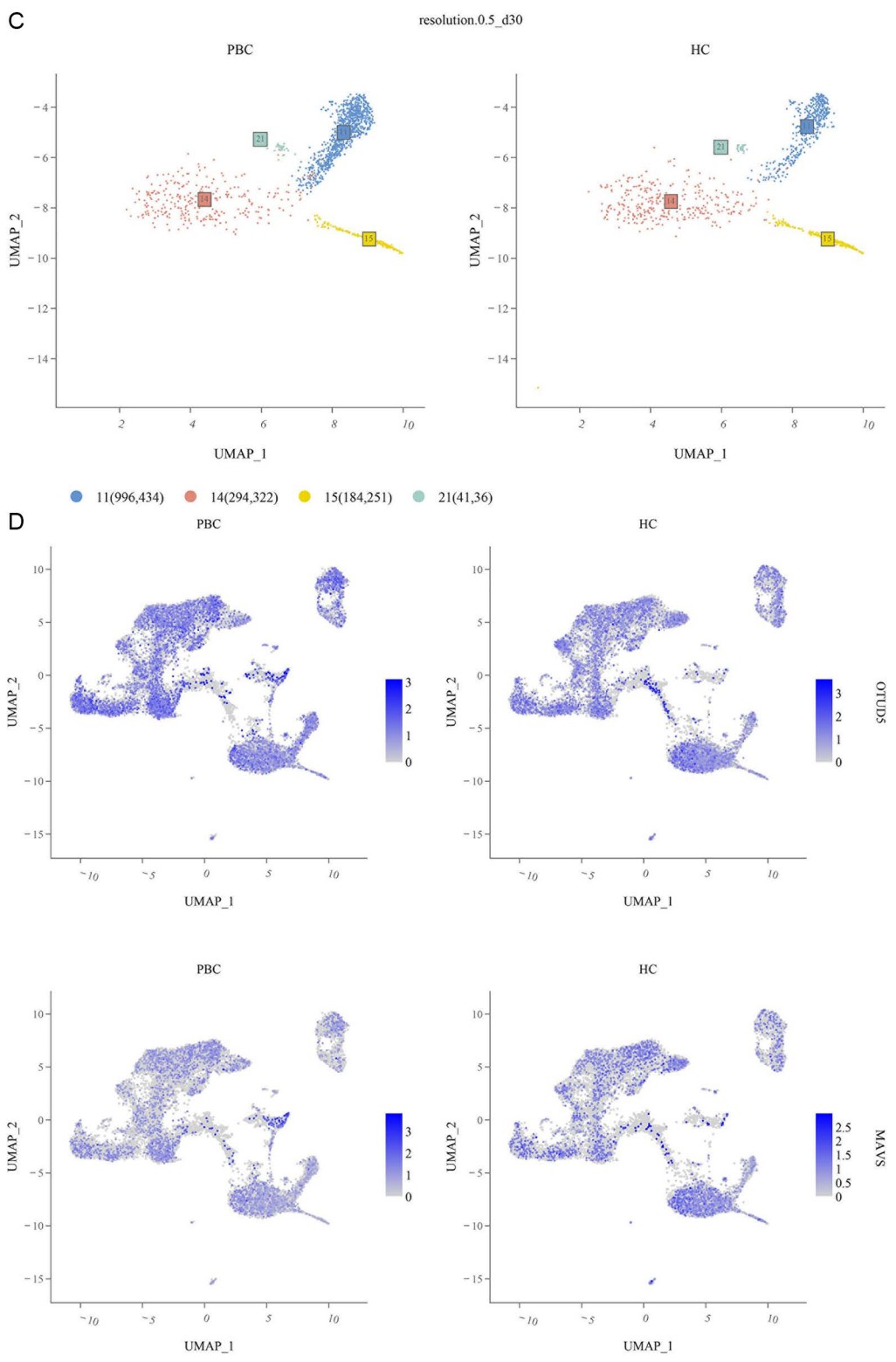

D

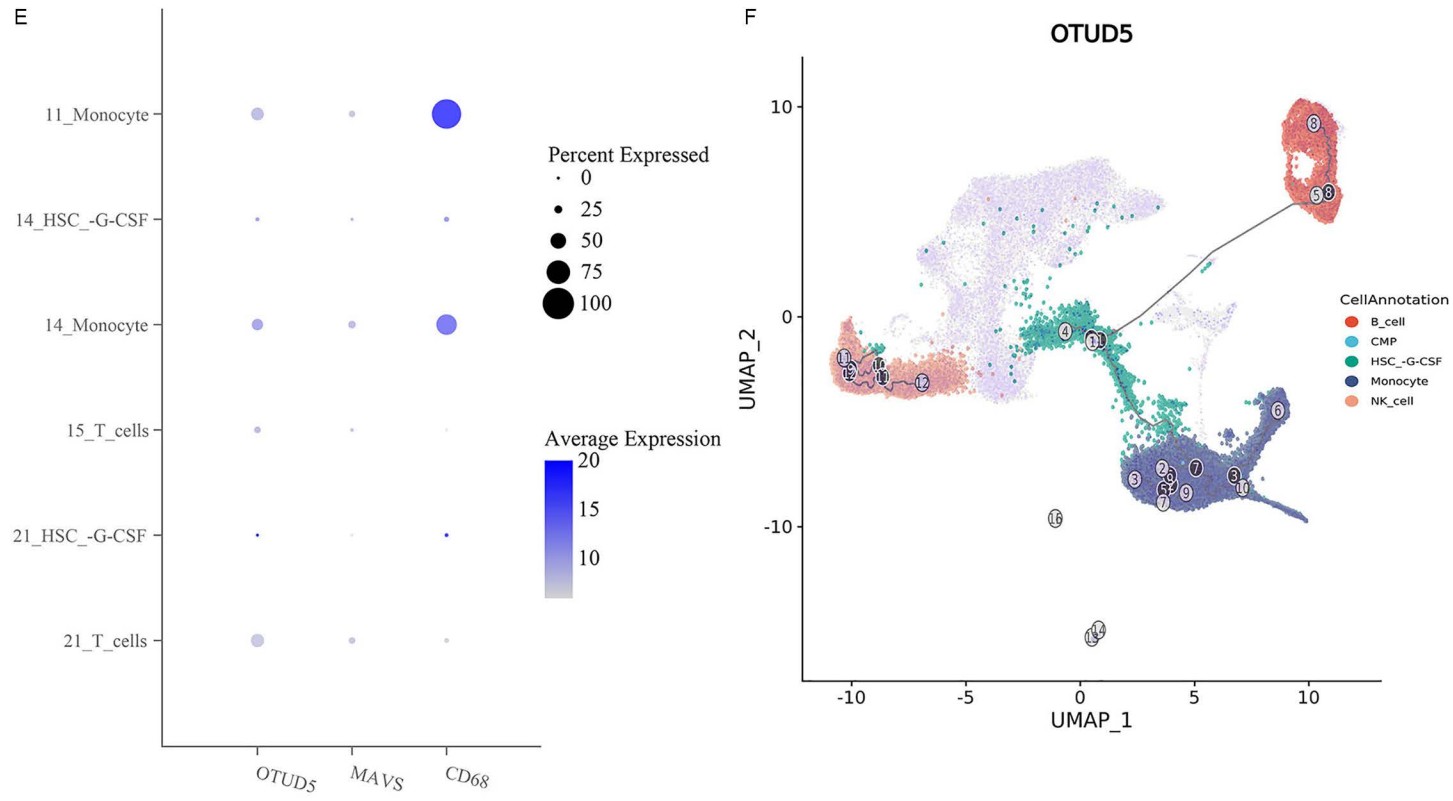

**Fig 8. PBC PBMCs single cell sequencing cell annotation results. A.** The classification annotation of major immune cells by PBC single cell sequencing, **B.** 22 cell subsets with a resolution of 0.5_d30 cell classification rule. **C.** The detailed classification analysis of monocytes, including subsets 11, 14, 15 and 21. **D.** The expression level of OTUD5 and MAVS in single-cell annotation of PBC and HC groups.The analysis results of differential gene expression in 11 mononuclear subsets. **E.** Dot size reflects the number of cells expressing differential genes. The bluer the color, the higher the average cell expression level. **F.** The simulated changes at different cell differentiation times with the accumulation of numbers in the pseudo time series.

OTUD5 has been previously linked to the antiviral response and apoptosis, whereas MAVS serves as a pivotal molecule in the antiviral immune response [31]. The interplay between these two factors may govern the immune response in patients with PBC, particularly in pathological macrophage subpopulations [32,33]. One study revealed that OTUD5 is prominently expressed in a range of autoimmune diseases, including systemic lupus erythematosus (SLE) and rheumatoid arthritis (RA) [34,35]. Furthermore, numerous studies have underscored the crucial role of MAVS in viral infection and immune cell activation [36,37]. Proteomic and immunofluorescence analyses revealed that the interaction between OTUD5 and MAVS could regulate the immunopathology of PBC. However, the downstream mitochondrial signalling pathways of OTUD5 and MAVS in PBC are unclear [37].

The results of the scRNA-seq and GEO analyses revealed a distinctive pattern of immune cells in PBC. This study identified biomarkers of OTUD5 and MAVS in a monocyte subpopulation of macrophages. This research also revealed the significance of sex-biased genes in patients with PBC. There were several specific molecules in subgroup 11 of PBC, including IFI30, TCF7L2, AIF1, and LST1, whose functions are related to MAPK binding and GTPase activity. M1-type macrophage differentiation relies on the MAPK and NF-κB signalling pathways [38,39]. While Xu et al. reported that

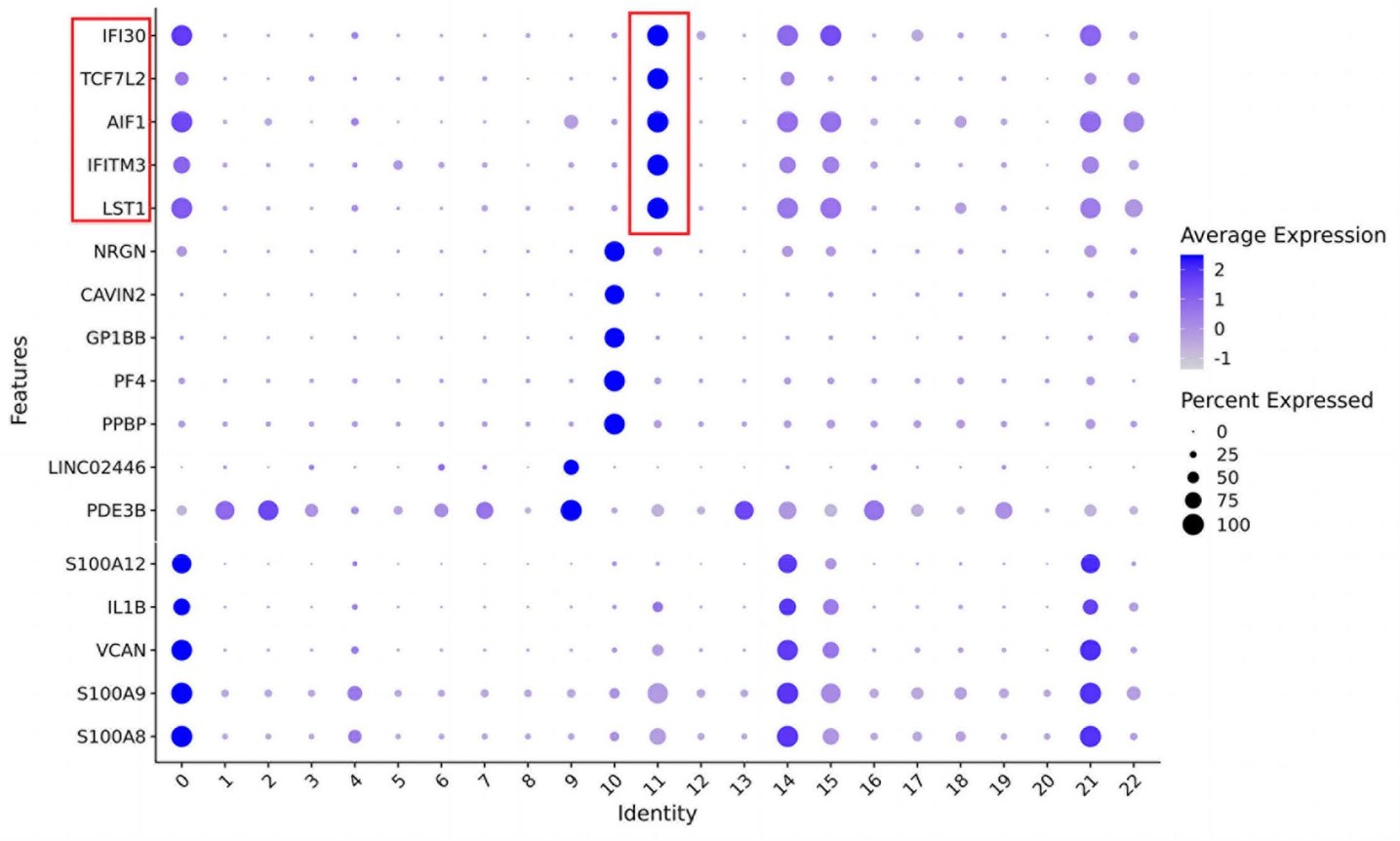

inflammatory traits are associated with NF-κB signalling in macrophage subsets of patients with PBC, they did not specify the specificity of monocyte subsets [38]. This study revealed the unique role of subgroup 11 monocytes in PBC, including increases in IFI30, TCF7L2, AIF1, and LST1. GSVA revealed high expression of IFN-γ and IFN-α, which are related to the inflammatory response in patients with PBC. This study revealed an effect of subgroup 11 on the CCR1 receptor and its cognate chemokines, such as CCL3, CCL3L1, and CCL5 [40,41]. This interplay could be crucial in the inflammatory cascade of PBC and merits further scrutiny. This study revealed the transcription factors of monocyte subset 11, namely, KLF4, MAFB, and RXRA, whose functions are related to inflammation and macrophage polarization; the importance of MAFB and RXRA in monocyte regulation and differentiation has also been reported [42–44]. However, the transcription factors of monocyte subset 11 involved in the inflammatory response and immune regulation in patients with PBC were elucidated for the first time in this work [45]. Most prior studies focused predominantly on the general functions of the MAPK and NF-κB signalling pathways, whereas this investigation focused predominantly on the high expression and interplay of MAVS and OTUD5 in monocyte subset 11 [46,47]. This finding offers a fresh perspective for understanding the cell-specific regulatory mechanisms underlying PBC. These findings have implications for advancing research and clinical practice in related bile duct diseases. We also explored the role of the X chromosome in immunity and PBC, deepening the understanding of biological phenomena and their underlying mechanisms, especially as integrating bioinformatics and immunogenetics may boost comprehension of intricate autoimmune liver diseases. Nevertheless, a primary limitation

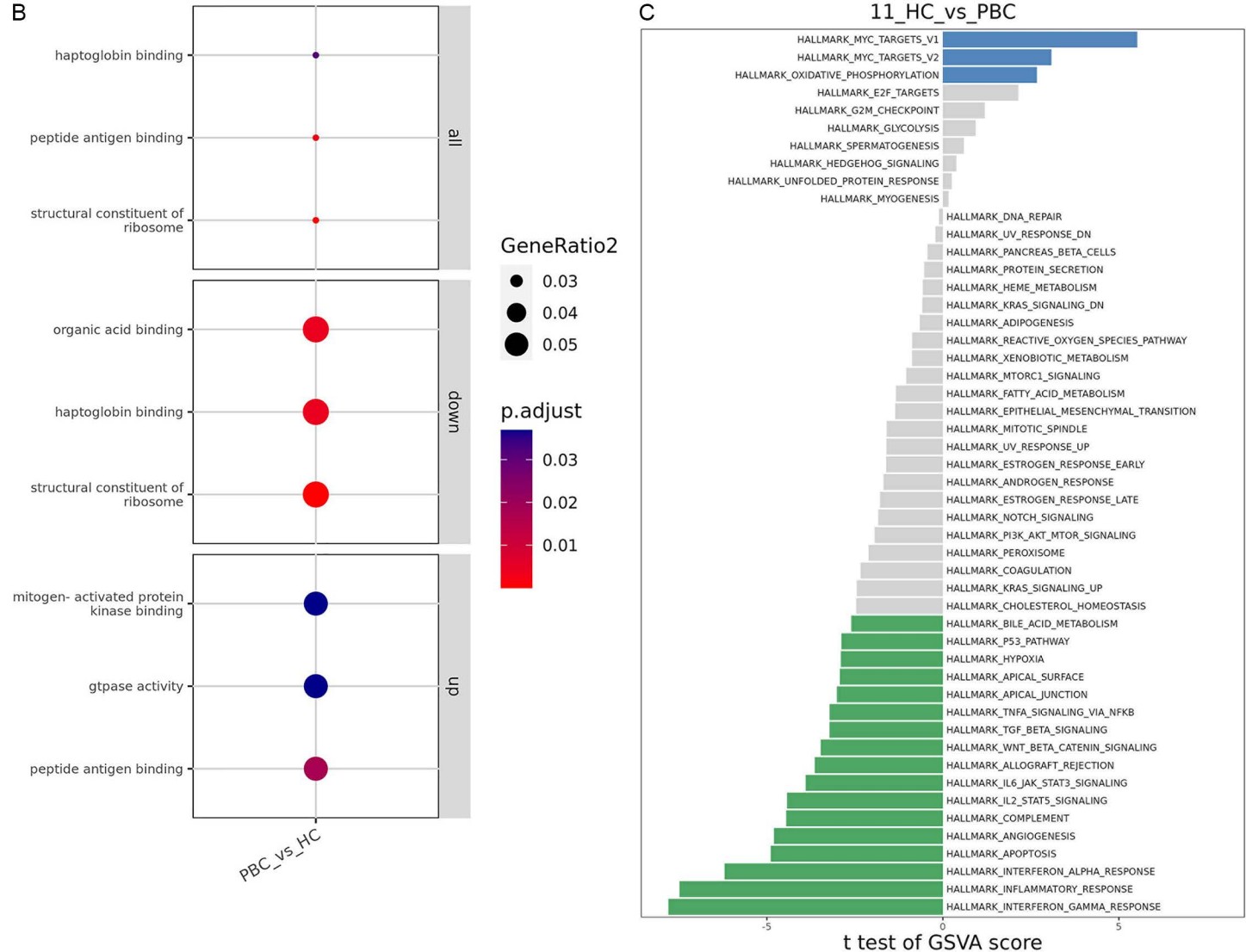

concerns the cohort size and sex distribution, highlighting the imperative for future research employing more extensive, larger sample sizes should be implemented in future research to validate these observations.

## 5. Conclusions

The gene expression profile and immune cell characteristics of patients with PBC were comprehensively analysed in this study. Through the use of advanced techniques such as machine learning and microarray data analysis, the increases in OTUD5, PQBP1 and TIMM17B were analysed in patients with PBC and those with PSC. Moreover, this study revealed an increase in the number of NK cells, Treg cells, and M0 macrophages and a decrease in the number of M2 macrophages in patients with PBC. This study revealed a potential interaction between OTUD5 and MAVS. This study also revealed a specific monocyte subset with OTUD5–MAVS interaction in PBC by single-cell sequencing. These findings provide new insights into the role of immune cells in PBC.

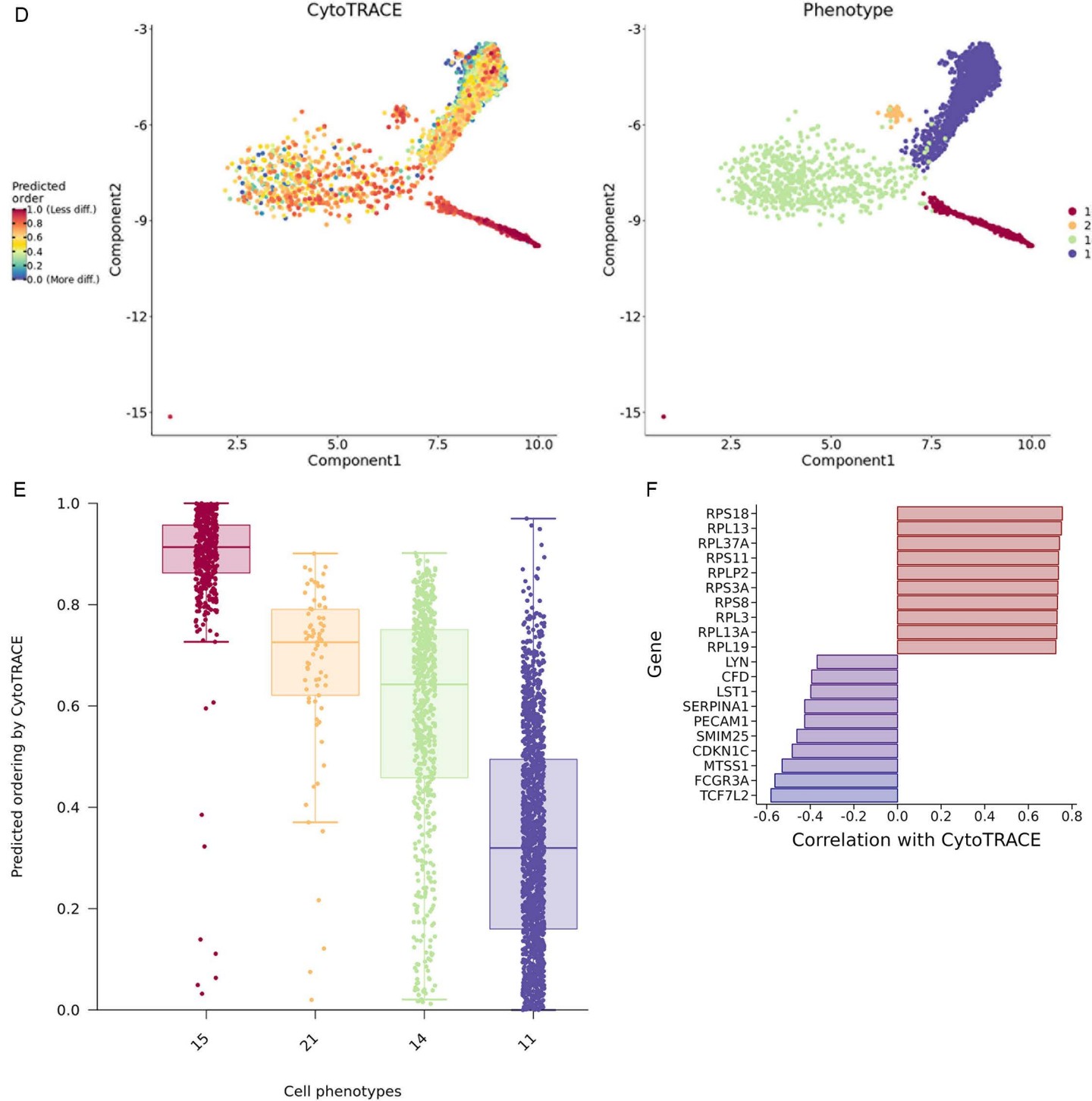

**Fig 9. Results of CytoTRACE differentiation of monocyte subsets 11 and enrichment of characteristic molecules. A.** The differentially expressed characteristic genes of subpopulation 11. **B.** GO enrichment for mono-11. **C.** GSEA functional enrichment results of gene clusters. **D.** The horizontal coordinate is cell type, and the vertical coordinate is CytoTRACE score. The higher the score, the lower the differentiation degree and the higher the dryness of the cell type. E. The phenotype of cell type, and the CytoTRACE score, the lower the score, the higher the differentiation. **F.** Expression of gene markers associated with high differentiation in 11 subpopulations.

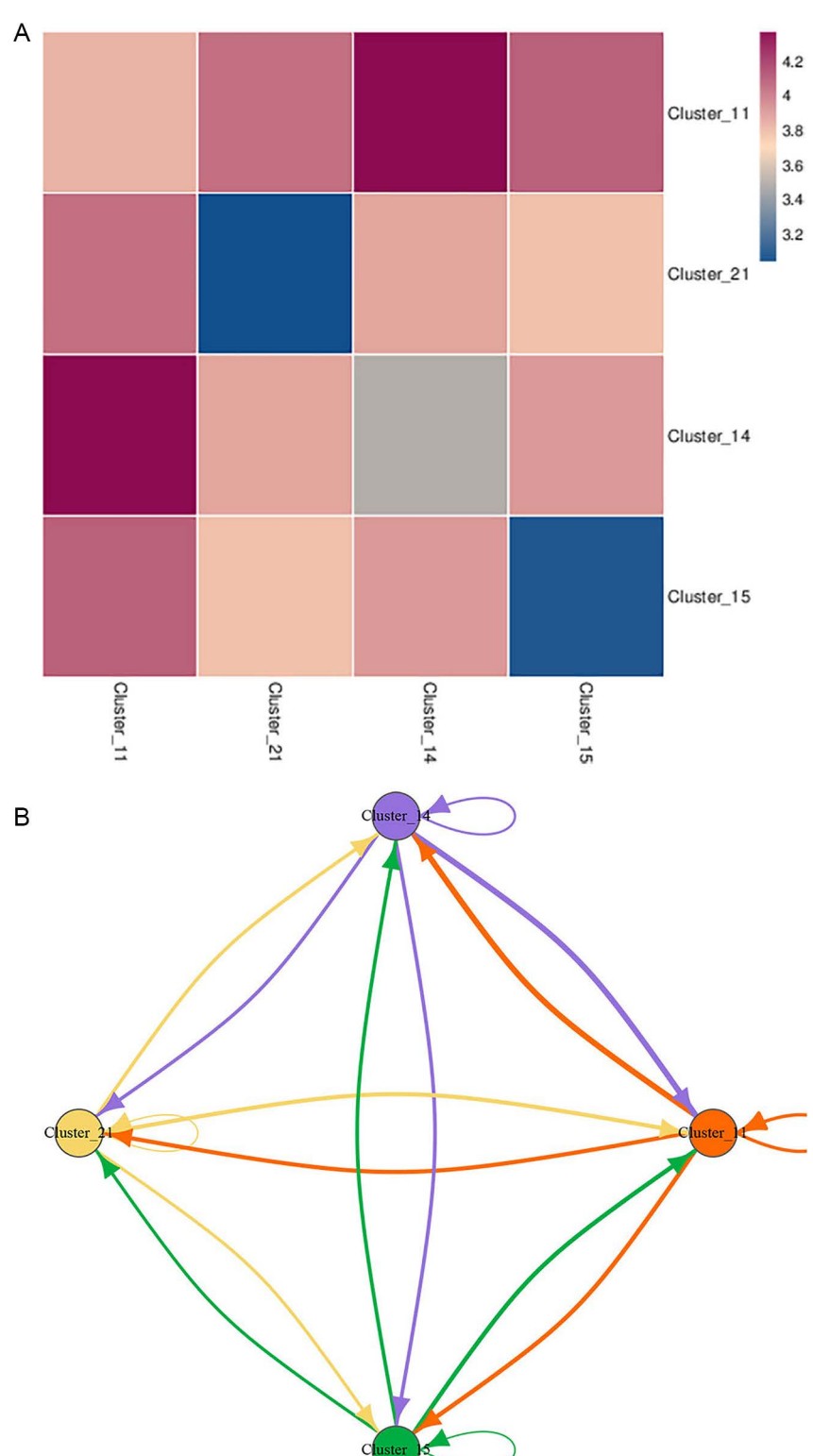

A

B

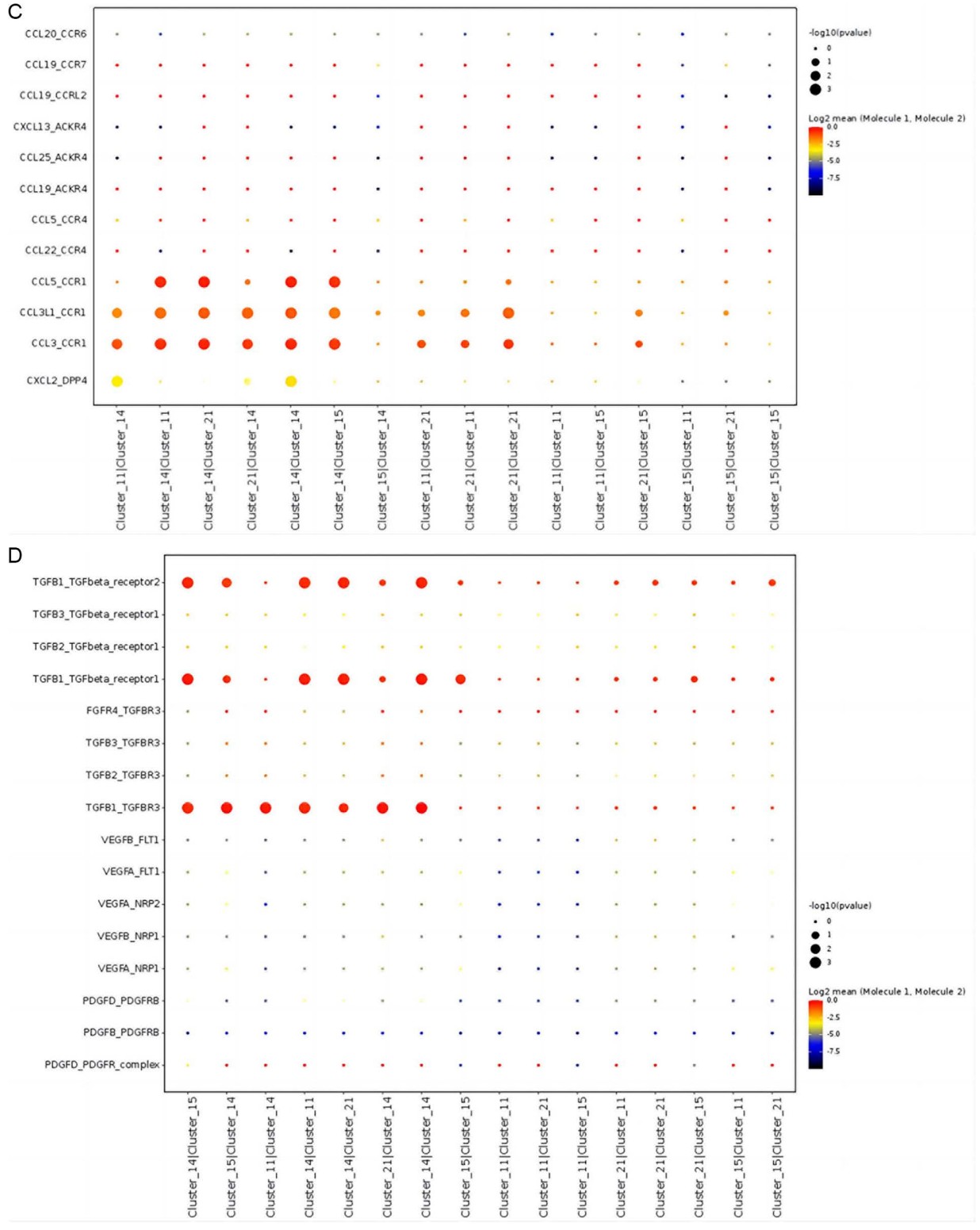

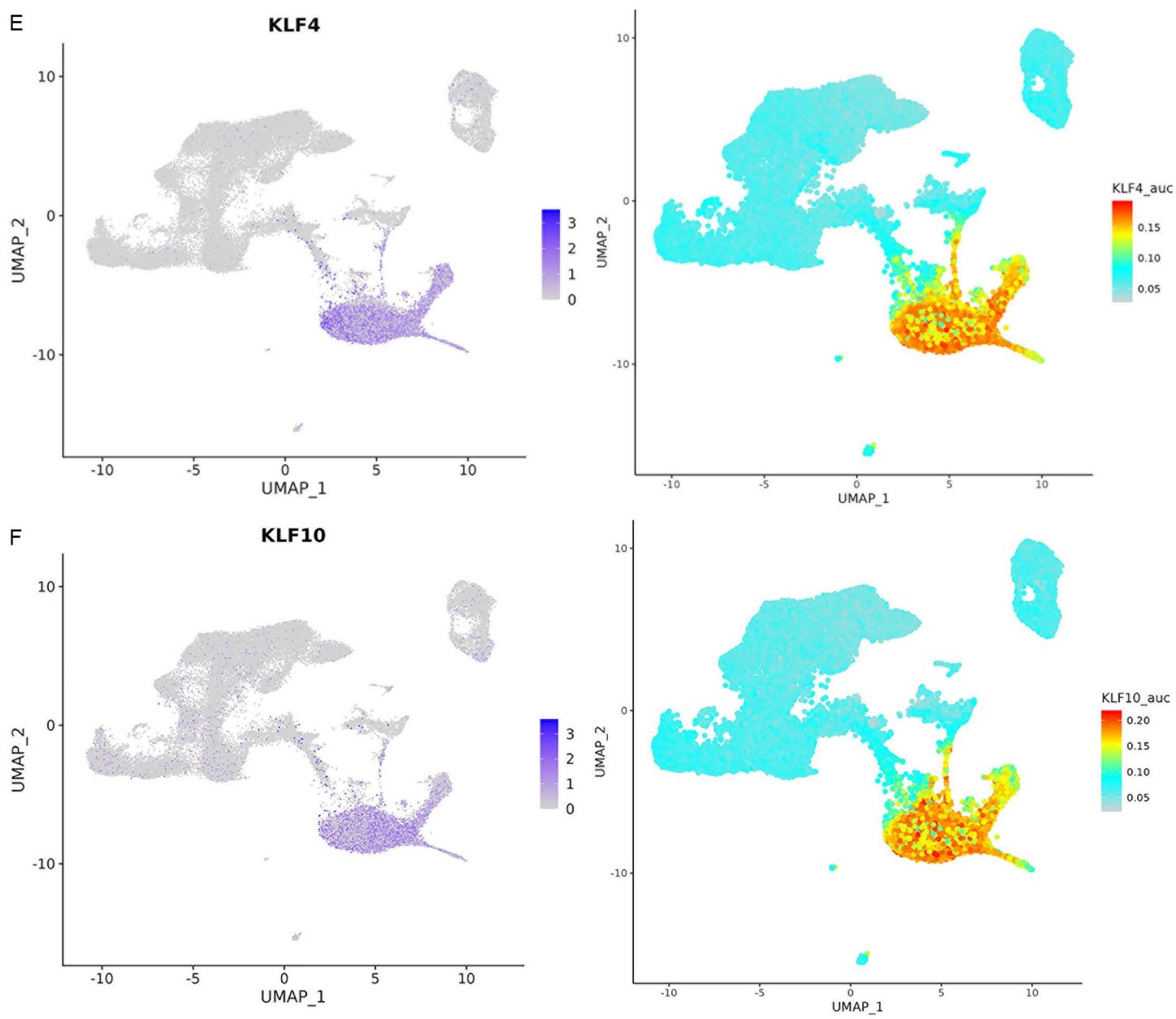

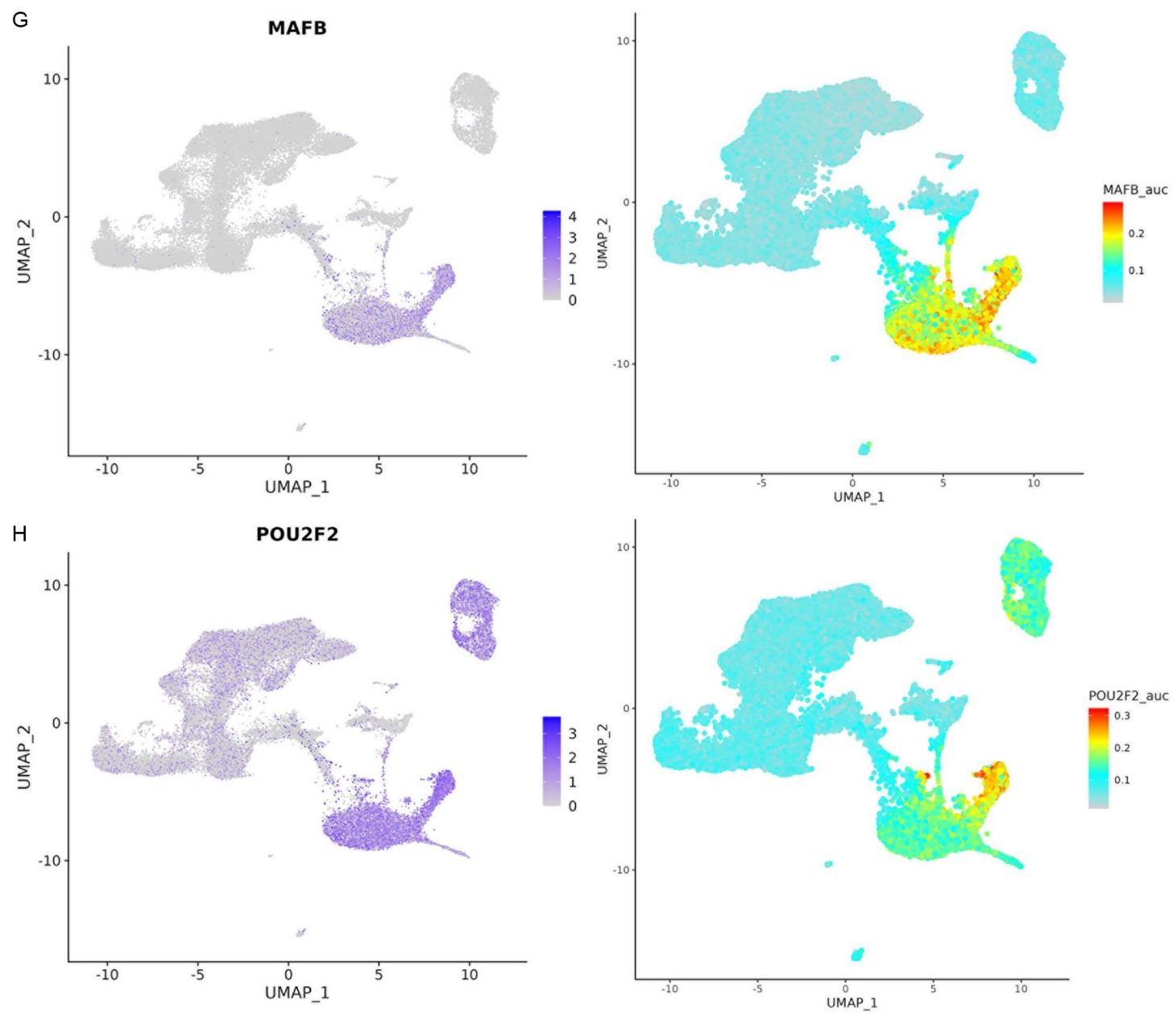

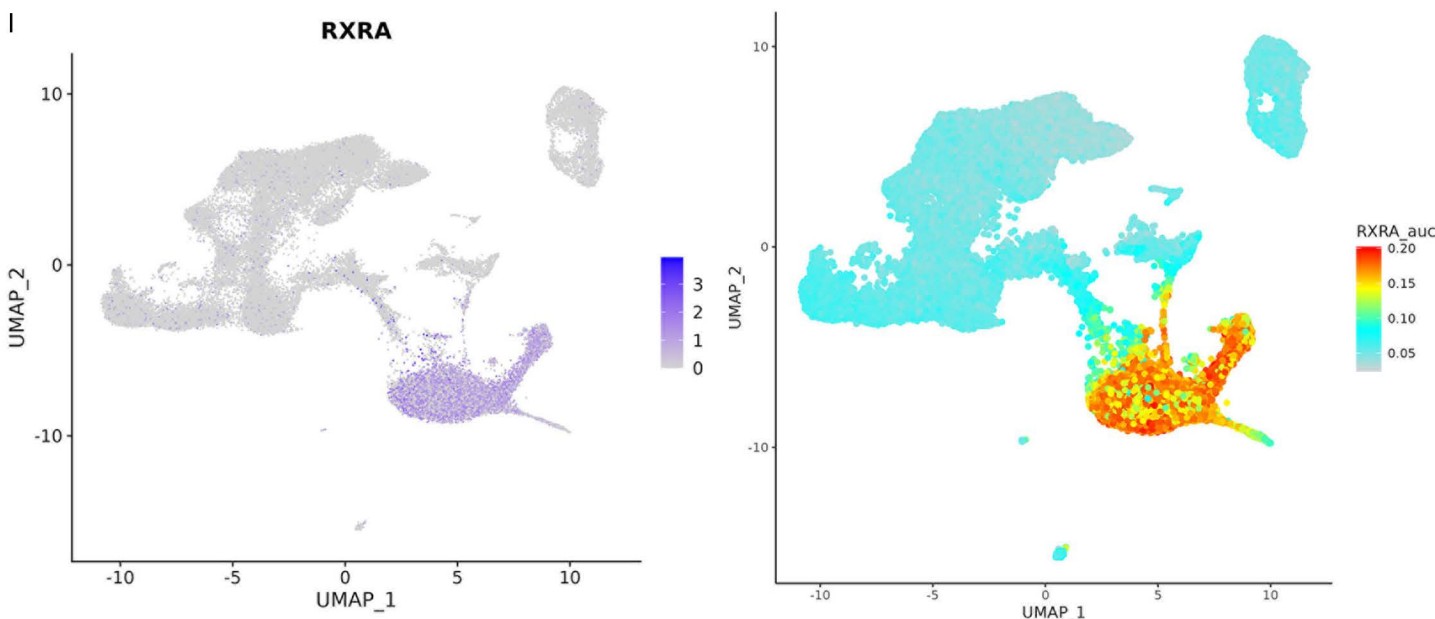

**Fig 10. Interactions between mononuclear 11 subpopulation and other related mononuclear derived differentiated cell communities and transcription factor interactions. A.** The interaction heat map shows that the redder the color, the more closely related the cell subsets are. The network diagram of direct interaction is established from the intensity of interaction. **B.** from the arrow direction and line thickness, it can be seen that subgroup 11 and subgroup 14 interact most closely. **C.** The larger the dot, the stronger the effect of the displacing factor. **D.** The larger the dot, the closer the effect. TGF-β1 and 3 receptors may be signaling pathways for differentiation between subgroups 11 and other subgroups. **E-I.** Distribution and expression characteristics of transcription factors among monocyte subsets. The higher the Auc score, the stronger the cell specificity of expression.

## Supporting information

**S1 Fig. By normalizing the data and limma processing, the differential genes were screened.** Filtering threshold as LogFC = 2. After adjustion, the P-threshold value is less than or equal to 0.05 (adj.P.Val ≤ 0.05). Heat maps show up-regulated genes in red and down-regulated genes in blue. **A.** Differentially expressed genes (DEGs) hubs were identified between PBC and HC control in samples of liver tissue.
(TIF)

**S2 Fig. By normalizing the data and limma processing, the differential genes were screened.** Filtering threshold as LogFC = 2. After adjustion, the P-threshold value is less than or equal to 0.05 (adj.P.Val ≤ 0.05). Heat maps show up-regulated genes in red and down-regulated genes in blue. **B.** Differentially expressed genes (DEGs) hubs were identified between PBC and HC control in whole blood.
(TIF)

**S3 Fig. By normalizing the data and limma processing, the differential genes were screened.** Filtering threshold as LogFC = 2. After adjustion, the P-threshold value is less than or equal to 0.05 (adj.P.Val ≤ 0.05). Heat maps show up-regulated genes in red and down-regulated genes in blue. **C.** Differentially expressed genes (DEGs) hubs were identified between PBC and HC control in whole blood.
(TIF)

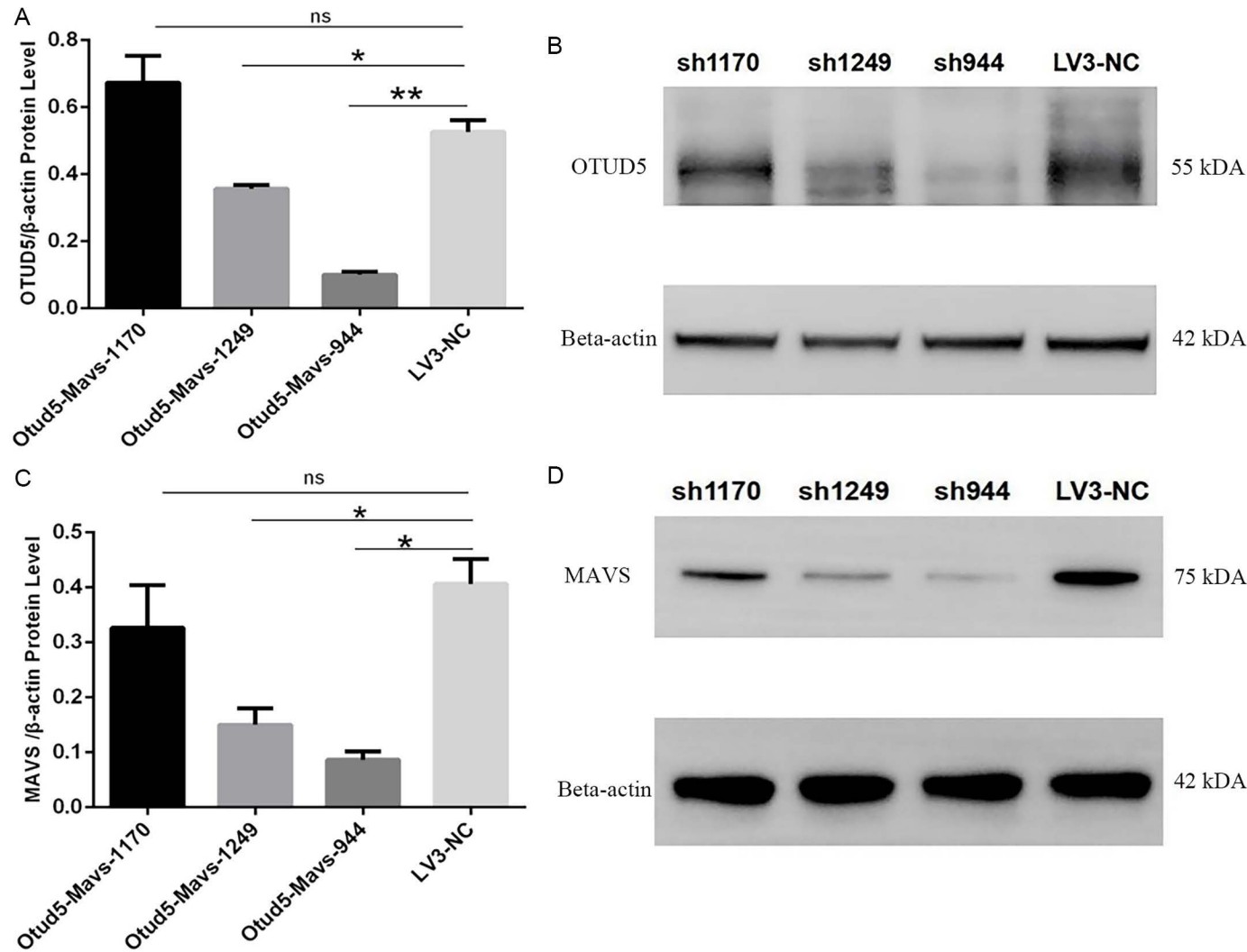

## Author contributions

**Conceptualization:** Yan Sun, Ran Chen, Wenlin Tai.

**Data curation:** Yan Sun, Ran Chen, Wenlin Tai.

**Formal analysis:** Yan Sun, Wenlin Tai.

**Funding acquisition:** Yan Sun, Ran Chen, Wenlin Tai.

**Investigation:** Ran Chen.

**Methodology:** Yan Sun, Ran Chen.

**Project administration:** Yan Sun, Ran Chen.

**Resources:** Ran Chen.

**Software:** Ran Chen.

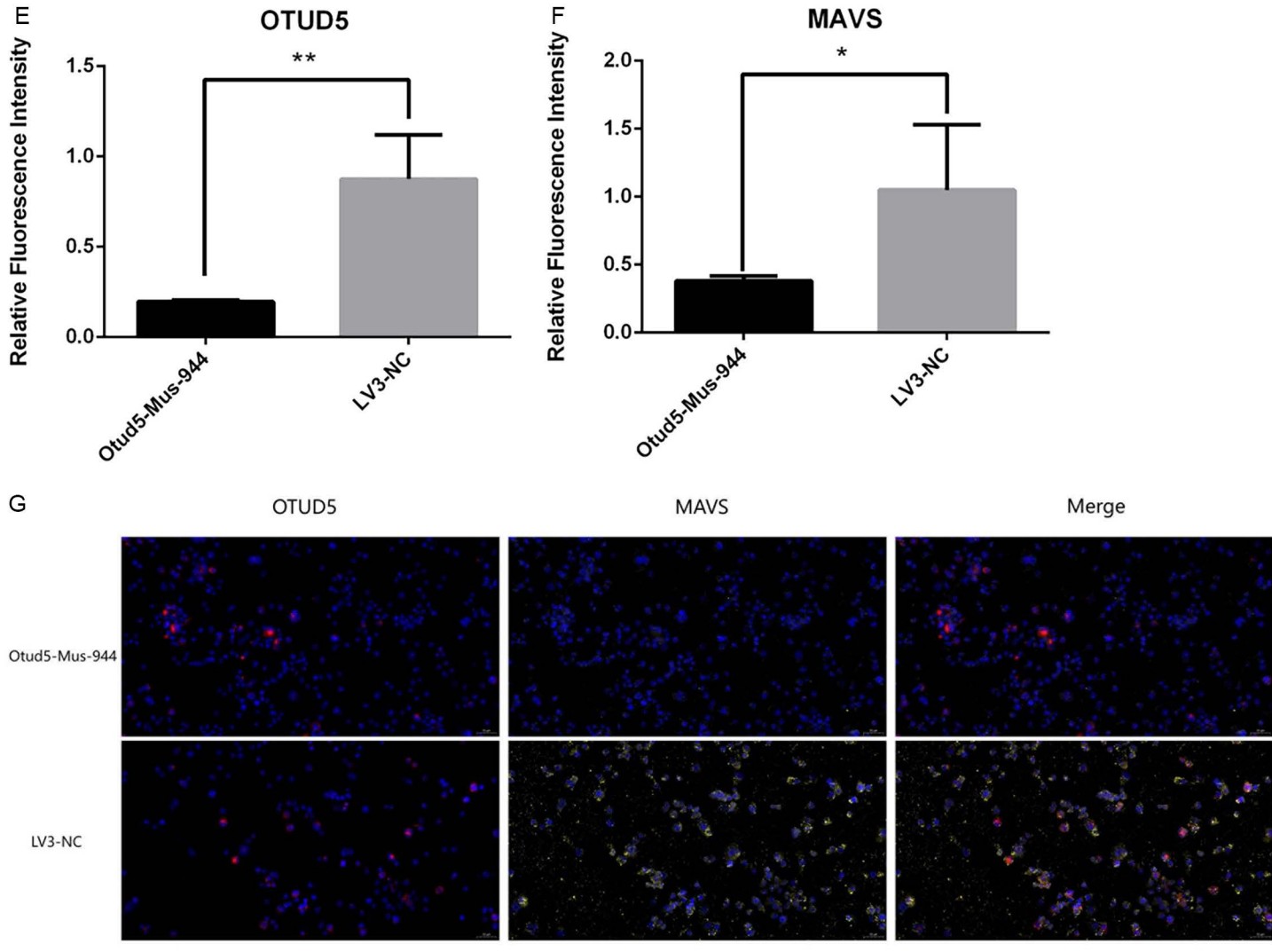

**Fig 11. The expression of OTUD5 knockdown RAW264.7 cell line was constructed. A&B.** The protein level and Western Blot of OTUD5-knockdown. **C&D.** the verification of MAVS after OTUD5 knockdown in RAW264.7. Immunofluorescence quantization of OTUD5 and MAVS protein after knockdown. **E&F.** The statistics of immunofluorescence respectively. *P*<0.05. **G.** The immunolocalization of OTUD5 and MAVS in RAW264.7 cells of OTUD5-MUS-944 and LV3-NC groups, Scar bar = 50 μm.

**Supervision:** Yan Sun, Ran Chen, Wenlin Tai.

**Validation:** Yan Sun, Ran Chen.

**Visualization:** Yan Sun, Ran Chen, Wenlin Tai.

**Writing – original draft:** Ran Chen.

**Writing – review & editing:** Yan Sun, Ran Chen, Wenlin Tai.

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
