## [Decision Letter · Decision Letter 0]

24 Sep 2025

PONE-D-24-53116Comparative study of Machine Learning associated with immune cell infiltration potentially for identifying significant genes in Primary Biliary CholangitisPLOS ONE

Dear Dr. Sun,

Thank you for submitting your manuscript to PLOS ONE. After careful consideration, we feel that it has merit but does not fully meet PLOS ONE’s publication criteria as it currently stands. Therefore, we invite you to submit a revised version of the manuscript that addresses the points raised during the review process.

We look forward to receiving your revised manuscript.

Kind regards,

Muhammad Salman Bashir, M.S.C

Academic Editor

PLOS ONE

5. Please include a copy of Table 1 which you refer to in your text on page 31.

Additional Editor Comments:

Reviewer #1:

Reviewer #2:

Reviewer #3:

Reviewers' comments:

Reviewer's Responses to Questions

**Comments to the Author**

1. Is the manuscript technically sound, and do the data support the conclusions?

Reviewer #1: Yes

Reviewer #2: Yes

Reviewer #3: Yes

2. Has the statistical analysis been performed appropriately and rigorously?

Reviewer #1: Yes

Reviewer #2: Yes

Reviewer #3: I Don't Know

3. Have the authors made all data underlying the findings in their manuscript fully available?

Reviewer #1: Yes

Reviewer #2: Yes

Reviewer #3: Yes

4. Is the manuscript presented in an intelligible fashion and written in standard English?

Reviewer #1: Yes

Reviewer #2: Yes

Reviewer #3: Yes

5. Review Comments to the Author

Reviewer #1: This is well written document mainly focusing on statistical data achievement using machine learning and in advancement to current LLM be trained on advancement of such studies.

The manuscript presents a comparative study of various machine learning (ML) algorithms applied to immune cell infiltration profiles with the goal of identifying significant genes in Primary Biliary Cholangitis (PBC). The topic is timely and relevant, particularly as integrating bioinformatics and immunogenetics can enhance understanding of complex autoimmune liver diseases. However, the study would benefit from improvements in several key areas, both scientifically and structurally.

Reviewer #2: The research is written in standard english. The topic is also very interesting. The manuscript is technically sound. The analysis is also done properly.the authors made all data underlying the findings in their manuscript fully available

Reviewer #3: When writing an abstract, the problem should be stated before the research study itself.

The discussion section should be separated from previous work, and the previous work should be placed after the introduction to enable comparison of their work with the current study.

6. PLOS authors have the option to publish the peer review history of their article (what does this mean?). If published, this will include your full peer review and any attached files.

Reviewer #1: **Yes:** Murad Zaheer Yousuf

Reviewer #2: No

Reviewer #3: No

---

## [Author Response · Author response to Decision Letter 1]

26 Sep 2025

Submission ID: PONE-D-24-53116

Title: "Comparative study of Machine Learning associated with immune cell infiltration potentially for identifying significant genes in Primary Biliary Cholangitis”

Correspondence Author: Yan Sun

We sincerely appreciate your thorough review of our manuscript and the invaluable, constructive feedback provided. We are confident that these revisions have substantially enhanced the manuscript's quality. We have meticulously addressed all comments and diligently incorporated the suggested modifications to the best of our ability. Below is our point-by-point response to your specific comments and the actions we have taken. The comments are laid out below in italicized font and specific concerns have been numbered, Our response is given in red font.

Please find the following Response to the comments of editor:

Editor comments:

1.Please ensure that your manuscript meets PLOS ONE's style requirements, including those for file naming. The PLOS ONE style templates.

2. Please note that PLOS ONE has specific guidelines on code sharing for submissions in which author-generated code underpins the findings in the manuscript. In these cases, we expect all author-generated code to be made available without restrictions upon publication of the work.

3. We note that you have indicated that there are restrictions to data sharing for this study. For studies involving human research participant data or other sensitive data, we encourage authors to share de-identified or anonymized data. However, when data cannot be publicly shared for ethical reasons, we allow authors to make their data sets available upon request. For information on unacceptable data access restrictions,

5. Please include a copy of Table 1 which you refer to in your text on page 31.

Response

We also extend our sincere gratitude to the editor for the careful and thorough attention given to our manuscript. Your feedback has been invaluable in helping us improve the quality of our work.

In accordance with your instructions, we have performed a comprehensive check and confirm that all journal requirements have been met in the revised manuscript.

The table has been inserted into the document and the availability of data has been refreshed according to the editor’ comment.

Response to the editor comments:

Reviewer #1:

This is well written document mainly focusing on statistical data achievement using machine learning and in advancement to current LLM be trained on advancement of such studies.

The manuscript presents a comparative study of various machine learning (ML) algorithms applied to immune cell infiltration profiles with the goal of identifying significant genes in Primary Biliary Cholangitis (PBC). The topic is timely and relevant, particularly as integrating bioinformatics and immunogenetics can enhance understanding of complex autoimmune liver diseases. However, the study would benefit from improvements in several key areas, both scientifically and structurally.

Response

Dear Murad Zaheer Yousuf,

I would like to express my sincere gratitude to you for their recognition of our work. It is a genuine honor to know that our efforts have been appreciated, and receiving such thoughtful and encouraging feedback feels like a ray of sunshine breaking through the clouds. In addition, at the conclusion of the Discussion, we have acknowledged the limitations of our study and proposed potential avenues for future research, which we hope will provide feasible directions for investigators in related fields. As detailed in the Discussion section (Page 43, Lines 674-678).

Reviewer #2:

The research is written in standard english. The topic is also very interesting. The manuscript is technically sound. The analysis is also done properly.the authors made all data underlying the findings in their manuscript fully available.

Response

We are sincerely grateful for your positive and encouraging comments regarding our manuscript. Your recognition of the language, the interesting nature of the topic, the technical soundness, the proper analysis, and the data availability is highly motivating and greatly appreciated.

Reviewer #3:

When writing an abstract, the problem should be stated before the research study itself. The discussion section should be separated from previous work, and the previous work should be placed after the introduction to enable comparison of their work with the current study.

Response

We are grateful to the reviewer for this excellent suggestion, which has helped us improve the logical flow of the abstract. It is important to clarify that our study was directly inspired by the initial findings from XWAS study. Our primary contribution lies in the rigorous validation of these findings and their development into a novel scientific hypothesis, which constitutes the central aim and novelty of our work. Following this, we have revised it to now begin with a clear statement of the previous work in the Introduction (Page 5, Lines 124-128). In addition, this revised structure (Background -> Methods -> Results -> Conclusion) indeed enhances readability. The updated abstract is presented in the abstract).

---

## [Decision Letter · Decision Letter 1]

20 Jan 2026

A multifaceted analysis of OTUD5 integrated MAVS in innate immunity of Primary Biliary Cholangitis

PONE-D-24-53116R1

Dear Dr. Sun,

We’re pleased to inform you that your manuscript has been judged scientifically suitable for publication and will be formally accepted for publication once it meets all outstanding technical requirements.

Kind regards,

Muhammad Salman Bashir, M.S.C

Academic Editor

PLOS One

Additional Editor Comments (optional):

Reviewers' comments:

Reviewer's Responses to Questions

**Comments to the Author**

1. If the authors have adequately addressed your comments raised in a previous round of review and you feel that this manuscript is now acceptable for publication, you may indicate that here to bypass the “Comments to the Author” section, enter your conflict of interest statement in the “Confidential to Editor” section, and submit your "Accept" recommendation.

Reviewer #1: All comments have been addressed

2. Is the manuscript technically sound, and do the data support the conclusions?

Reviewer #1: Yes

3. Has the statistical analysis been performed appropriately and rigorously?

Reviewer #1: Yes

4. Have the authors made all data underlying the findings in their manuscript fully available?

Reviewer #1: Yes

5. Is the manuscript presented in an intelligible fashion and written in standard English?

Reviewer #1: Yes

6. Review Comments to the Author

Reviewer #1: Addressed the points where attention was drawn in a professional way. I would recommend to proceed with revised version.

7. PLOS authors have the option to publish the peer review history of their article (what does this mean?). If published, this will include your full peer review and any attached files.

Reviewer #1: **Yes:** Murad Zaheer Yousuf

---

## [Editor Report · Acceptance letter]

PONE-D-24-53116R1

PLOS One

Dear Dr. Sun,

I'm pleased to inform you that your manuscript has been deemed suitable for publication in PLOS One. Congratulations! Your manuscript is now being handed over to our production team.

Kind regards,

on behalf of

Dr. Muhammad Salman Bashir

Academic Editor

PLOS One